# Enabling Co-Innovation for a Successful Digital Transformation in Wind Energy Using a New Digital Ecosystem and a Fault Detection Case Study



Sarah Barber [1,*], Luiz Andre Moyses Lima [2], Yoshiaki Sakagami [3], Julian Quick [4], Effi Latiffianti [5,6], Yichao Liu [7], Riccardo Ferrari [8], Simon Letzgus [9], Xujie Zhang [10] and Florian Hammer [1]

1   Institute for Energy Technology, Eastern Switzerland University of Applied Sciences, Oberseestrasse 10, 8640 Rapperswil, Switzerland; florian.hammer@ost.ch
2   Voltalia, 84 bd de Sébastopol, 75003 Paris, France; l.lima@voltalia.com
3   Federal Institute of Santa Catarina, Av. Mauro Ramos 950, Florianópolis 88020-300, Brazil; yoshi@ifsc.edu.br
4   Turbulence and Energy Systems Laboratory, University of Colorado, Boulder, CO 80309, USA; julian.quick@colorado.edu
5   Department of Industrial and Systems Engineering, Texas A & M University, College Station, TX 77843, USA; latiffianti@tamu.edu
6   Institut Teknologi Sepuluh Nopember, Jawa Timur 60111, Surabaya, Indonesia
7   Electric Power Research Institute (EPRI) Europe, NexusUCD, Block 9 & 10 Belfield Office Park, Beech Hill Road, D04 V2N9 Dublin, Ireland; yliu1@epri.com
8   Delft Center for Systems and Control, Delft University of Technology, Mekelweg 2, 2628 CD Delft, The Netherlands; r.ferrari@tudelft.nl
9   Institute for Energy Technology, Technische Universität Berlin, Str. des 17. Juni 135, 10623 Berlin, Germany; simon.letzgus@tu-berlin.de
10   Faculty of Mechanical Engineering and Automation, Zhejiang Sci-Tech University, Hangzhou 310018, China; zhang_xj_97@foxmail.com
*   Correspondence: sarah.barber@ost.ch

**Abstract:** In the next decade, further digitalisation of the entire wind energy project lifecycle is expected to be a major driver for reducing project costs and risks. In this paper, a literature review on the challenges related to implementation of digitalisation in the wind energy industry is first carried out, showing that there is a strong need for new solutions that enable co-innovation within and between organisations. Therefore, a new collaboration method based on a digital ecosystem is developed and demonstrated. The method is centred around specific "challenges", which are defined by "challenge providers" within a topical "space" and made available to participants via a digital platform. The data required in order to solve a particular "challenge" are provided by the "challenge providers" under the confidentiality conditions they specify. The method is demonstrated via a case study, the EDP Wind Turbine Fault Detection Challenge. Six submitted solutions using diverse approaches are evaluated. Two of the solutions perform significantly better than EDP's existing solution in terms of Total Prediction Costs (saving up to €120,000). The digital ecosystem is found to be a promising solution for enabling co-innovation in wind energy in general, providing a number of tangible benefits for both challenge and solution providers.

**Keywords:** wind energy; digitalisation; collaboration; co-innovation; machine learning; fault detection

## 1. Introduction

The successful exploitation of the potential benefits of digitalisation is one of the key topics in the wind energy community today. Recently-formed international collaborations such as IEA Wind Task 43 (https://www.ieawindtask43.org/; accessed on 25 July 2022) and the WindEurope Digitalisation Taskforce aim to bring together members of the entire wind energy space in order to accelerate this process. Within WindEurope, the recent publication "Wind energy digitalisation towards 2030" concludes that the continued digitalisation of

wind farm construction, operation and maintenance (O&M) will be a major driver for reducing wind energy costs and risks in the next decade (https://windeurope.org/intelligence-platform/product/wind-energy-digitalisation-towards-2030/; accessed on 25 July 2022). Some results of the work within IEA Wind Task 43 include the collaborative paper "Grand Challenges in the Digitalisation of Wind Energy" [1], the Wind Resource Assessment (WRA) Data Model (https://github.com/IEA-Task-43/digital_wra_data_standard; accessed on 25 July 2022) and the definition of actions required to improve data commonality in wind energy [2]. The "Grand Challenges in the Digitalisation of Wind Energy" identified the following three Grand Challenges of wind energy digitalisation: (1) Creating FAIR data frameworks (FAIR: findable, accessible, interoperable and reusable [3]; (2) Connecting people and data to foster innovation; (3) Enabling collaboration and competition between organisations. Solutions to these Grand Challenges have already been investigated to some extent, as described below.

### 1.1. Creating FAIR Data Frameworks

FAIR data frameworks are findable, accessible, interoperable and reusable [3], where "findable" means it can be discovered by people or machines using a search engine, "accessible" means that it needs to be retrievable using secure but open and free protocols, for example through the internet, "interoperable" means that it can be used in workflows and/or applications, i.e., that the data and its relationships are machine-readable, and "reusable" means that it can be applied to different settings. Data can be made findable by the use of metadata, which describes the data and follows a defined schema.

Several efforts have already been made to encourage researchers to adopt FAIR principles. This includes an initiative by the European Commission to partly assess research funding applications according to their plans to provide open access to data and publications, and indirect funding of the development of sector-specific taxonomies. For example, the Sharewind metadata registry was created by the members of the European Energy Research Alliance Joint Programme on Wind Energy (EERA JP Wind Energy) as part of the European FP7 Coordination Action project IRPWind [4]. In addition to this, the topic of "open science" has been officially adopted by the new "Horizon Europe" funding scheme (https://ec.europa.eu/info/research-and-innovation/strategy/strategy-2020-2024/our-digital-future; accessed on 25 July 2022). In the United States, the Department of Energy released a project funding call in 2020 on the topic of artificial intelligence frameworks that utilise FAIR principles (https://www.energy.gov/articles/department-energy-announces-85-million-fair-data-advance-artificial-intelligence-science; accessed on 25 July 2022).

Despite this progress, many barriers must still be overcome, including making research data findable, and making data from the industry available, by, for example, solving the problem related to the fear of losing competitive advantage. The wind energy community is attempting to do this by, for example, developing a set of standard metadata with specific taxonomies (https://www.wedowind.ch/task-43-space; accessed on 25 July 2022) and by further developing the web-based data registry ShareWind.eu, allowing tagging of research data to assign a DOI to datasets in order to improve citability. In addition to this, data quality is key to ensuring interoperability and reusability of data [5]. A low data quality, which includes issues such as inconsistencies, inaccuracies, missing data and lack of metadata, was reported as one of the main barriers to data sharing in the interviews carried out in [1]. This not only requires data standards, improved measurement equipment, automated data filling algorithms and consistency checks, but also standardised and high quality filtering and quality control methods [5].

However, a centralised location specifically for a particular industry sector, such as wind energy, where all the available data are summarised, described and accessible (if relevant), is not known to the authors. This topic is taken into account in the development of the collaboration method in this present paper (see Section 2).

### 1.2. Connecting People and Data to Foster Innovation

Previous work on the second Grand Challenge "connecting people and data to foster innovation" includes work on internal company culture, on data-driven innovation as well as on methods to incentivise data and knowledge sharing.

Regarding internal company culture, it has been shown that organisations need the following things to foster innovation: (a) effective communication channels to spread ideas across the organisation, (b) a culture which allows people to speak out openly, (c) leadership that fosters critical thinking, and (d) autonomy that allows every employee to act [6]. In addition to this, digitalisation strategies setting out "a commitment to a set of coherent, mutually reinforcing policies or behaviours aimed at achieving a specific competitive goal" have been found to be valuable for exploiting the opportunities of digitalisation [7]. However, the application of these strategies to the wind energy industry has not been discussed in the literature.

On the topic on data-driven innovation, data are becoming increasingly important for innovation and co-innovation processes within organisations. Many companies, including Barilla, Twitter and Deliveroo, use digital platforms intensively to collect data from interactions with their stakeholders and leverage it for their internal innovation processes [8]. The subject of "co-innovation" refers to the process of exchanging ideas and resources via any type of physical or digital collaborative channels, involving all types of stakeholders (e.g., [9]). It enables people and organisations to use modern digital technologies for integrating and exchanging knowledge, ideas, resources and information. It is becoming increasingly popular due to its proven ability to solve the type of systemic, multidisciplinary, multi-stakeholder problem involving Big Data typical to today's challenges [10]. The only published work on the application of "co-innovation" concepts to energy transition challenges to the authors' knowledge relates to a case study of Japan and China demonstrating that technology-supplying countries and technology-importing countries can both benefit by co-innovating products [11]. There certainly appears to be a potential gap in experience with "co-innovation" in the wind energy industry.

Another method of connecting people and data to foster innovation is to focus on the needs and desires of the people who are supposed to be doing the data sharing. The results of a survey about the barriers of data sharing carried out as part of IEA Wind Task 43 show that, as well as making data FAIR, people need a real and tangible incentive in order to share data (and knowledge) [1]. This task is challenging due to the number of different stakeholders with different needs. For example, a data scientist might be a strong supporter of sharing data and knowledge in order to learn from others but may work for a company who prevents them from doing so due to legal, structural or policy reasons. Some recent initiatives to incentivise data sharing include data marketplaces such as the Greenbyte marketplace for wind data (https://www.greenbyte.com/marketplace/wind-only; accessed on 25 July 2022) and the IntelStor Market Intelligence Ecosystem (https://www.intelstor.com/; accessed on 25 July 2022), data discovery and sharing platforms such as the Sharewind metadata catalogue (https://sharewind.eu/; accessed on 25 July 2022) and the US DOE Data Archive & Portal (https://a2e.energy.gov/about/dap; accessed on 25 July 2022), comparison and benchmarking activities such as IEA Wind Task 31 (https://iea-wind.org/task31/; accessed on 25 July 2022), IEA Wind Task 30 (OC6) WP3 Benchmark (https://iea-wind.org/task30/) and CREYAP: Comparison of Resource and Energy Yield Assessment Procedures [12], and challenge-based platforms such as Kaggle (https://www.kaggle.com/; accessed on 25 July 2022), Knowledge Pit [13] and the EDP Open Data Platform (https://opendata.edp.com/; accessed on 25 July 2022). To the authors' knowledge, there is no scientific literature that compares or evaluates these different initiatives.

However, literature on the general topic of incentivising data and knowledge sharing exists. A recent review on incentivising research data summarises the main requirements for incentivising researchers to share data [14]. These include: (a) build on existing cultures and practices, (b) meet people where they are and tailor interventions to support them, (c)

promote disciplinary data champions to model good practice and drive cultural change, (d) provide robust technical infrastructure and protocols, such as labelling of data sets, data standards and use of data repositories.

For wind energy in particular, the topic has been investigated as part of IEA Wind Task 43. Several interviews have been carried out regarding data sharing and sharing incentives, as described in [1]. One of the findings was that making open-source tools available and encouraging their use can incentivise data sharing because the shared data get used and its added value becomes more clear. Existing open-source tools in wind energy include the Brightdata app (https://www.brightwindanalysis.com/brightdata/; accessed on 25 July 2022), OpenOA (https://github.com/NREL/OpenOA; accessed on 25 July 2022) and the Data Science for Wind Energy R Library (https://github.com/TAMU-AML/DSWE-Package/; accessed on 25 July 2022). Existing initiatives to develop open data standards include the already mentioned IEA Wind Task 43 WRA Data Model and Metadata Challenge, as well as the ENTR Alliance (https://www.entralliance.com/; accessed on 25 July 2022).

Although there are many promising activities underway, there is limited experience about how the results actually benefit the different stakeholders in the industry. This present paper contributes to closing this gap.

*1.3. Enabling Collaboration and Competition between Organisations*

Previous work on the third Grand Challenge, "Enabling collaboration and competition between organisations", includes both co-innovation and fair evaluation methods.

Organisations are increasingly sharing data with various partners for collaborative innovation purposes [8]. "Collaborative innovation" focuses on the development of collaborative networks between organisations, and involves sharing knowledge, experience and resources in order to develop collaborative innovations, for example by creating structured partnerships and alliances [15]. As well as "collaborative innovation", the concepts of "open innovation" and "co-creation" are commonly used in this context. "Open innovation" refers to the acquisition of knowledge and resources from external partners, whereas "co-creation" refers to the involvement of customers in companies' product and service innovation processes. In fact, the concept of "co-innovation" introduced in Section 1.2 is positioned at the intersection of "collaborative innovation", "open innovation" and "co-creation". It is therefore not only applied within companies but also to enable collaboration and competition between organisations. "Co-innovation" concepts are becoming increasingly useful and popular due to the recent reduction in costs and increased availability of web-based technologies. They not only accelerate the processes of knowledge creation and sharing (e.g., [15,16]) but allow the development of specific digital interaction platforms through which flexible and dynamic "co-innovation" processes can be implemented via combination with physical collaboration channels (e.g., [17]). Although several data sharing, open data and challenge-based platforms already exist, as described in Section 1.2, there seems to be a high potential for the application of "co-innovation" methods in the wind energy sector in order to improve collaboration and competition between organisations.

Although the concept of "co-innovation" has not yet been applied in the wind energy sector, the idea of "co-creation" has received recent attention in order to improve the acceptance of wind energy projects in local communities. For example, the links between "co-creation" and wind energy development were investigated, showing that new roles for citizens as co-creators and co-producers of electricity and planning decisions are needed [18]. A further study into the idea of treating citizens as co-producers of wind energy characterised public engagement into three types of co-production: (1) Local co-production, in spatially proximate wind energy projects; (2) Collective co-production, performed through collaboration among different actors in the wind energy sector, joined ownership or consumption of wind energy; (3) Virtual co-production, mediated through information technology [19]. These studies should also be considered in the development of a solution used to enable collaboration and competition between organisations.

Competition between organisations also has an important role to play in exploiting the full potential of digitalisation in wind energy. In order for competition to be used effectively to further the industry as a whole, the results need to be comparable in a fair and agreed-upon way. This poses new challenges. Within the wind energy sector, some experience has been gained on this topic via the benchmarking projects introduced in Section 1.2. Specifically, the benchmarking within IEA Wind Task 31 has led to the development of the Wind Energy Model Evaluation Protocol (https://wemep.readthedocs.io/en/latest/index.html; accessed on 25 July 2022), which provides open-source documentation on model evaluation procedures and quality-checked verification and validation benchmarks for wind resource assessment. In addition to this, the CREYAP project has led to recommendations for future comparison end benchmarking projects and tools. Evaluation procedures can also be found on the EDP Open Data Platform (https://opendata.edp.com/; accessed on 25 July 2022).

Informal comparisons of prediction models for wind energy applications have been carried out by evaluating the prediction error in terms of Mean Absolute Error (MAE), Maximum Absolute Error (MAXAE), Root Mean Square Error (RMSE) and correlation coefficient (e.g., [20]). However, to the authors' knowledge, no systematic study has been carried out aimed at identifying relevant state-of-the-art evaluation methods that could be applied to enhance collaboration and competition between organisations. Possible methods that should be further investigated include Explainable Recommender Systems, which have been suggested as a potential way of building fair and transparent tools for evaluating Machine Learning (ML) models and work by conveying the reasoning behind its predictions [21], as well as decision support methods such as the recent introduction of methods for the comparison and evaluation of Artificial Intelligence (AI) tools in a fair, transparent and explainable way [22]. These methods seem to have a high potential for future application to the wind sector.

*1.4. Goals of the Present Work*

The literature review has shown that a focused effort on overcoming the barriers to a successful digital transformation of the wind energy sector is required. In order to initiate this process, a new collaboration method that has the potential to make wind energy data FAIR, enable co-innovation within and between organisations, incentivise data sharing and allow a fair evaluation of solutions is presented in this paper, together with a real case study. The paper is focused on the overall ecosystem and its potential for addressing the challenges of digitalisation, rather than on one of the specific aspects, which require further investigation. The collaboration method is introduced in Section 2.1 and the case study is described in Section 3.3. This is followed by a discussion of the results and an evaluation of the new method in Section 4, as well as a discussion on the future development and application of the method.

## 2. The Collaboration Method

*2.1. Requirements*

As well as the requirements from the literature review summarised at the end of Section 1, further requirements were defined based on the results of the IEA Wind Task 43 survey presented in the "Grand Challenges in the Digitalisation of Wind Energy" paper [1]. The most important aspects named by the 30 interviewed members of the global wind energy sector in order to improve data sharing were:

- Owner/operators: Getting all the data in one spot; IT issues; Cleaning/filtering raw data (different time scales and resolutions, different formats); Refining and processing data ready for machine learning model (80% of time); Interfaces to collect data reliably;
- Academia: Lack of public data; No standard format for analysing and processing data; Poor data quality; Lack of willingness to share data, especially higher resolution; Lack of change logs;

- Technology providers: Data quality; Different format and structure of data; Data filtering for analyses; Data collection: different devices need to be programmed differently; Time for downloading, cleaning, and training data.

This led to the conclusion that the new collaboration method should have the following characteristics:

- Enable co-innovation within and between organisations;
- Incentivise data sharing and allow a fair evaluation of solutions, with a particular focus on contextual and higher-frequency data;
- Make wind energy data FAIR (Findable, Accessible, Interoperable and Reusable);
- Provide a central location for data and knowledge related to a certain topic within the sector;
- Include solutions and code for data filtering and standard analysis tasks;
- Allow data standards and data structure translation solutions to be published and shared.

### 2.2. Method Description

The collaboration method developed here **enables flexible and dynamic co-innovation within and between organisations** by combining a digital challenge-based platform with moderated workshops within a digital ecosystem. As shown in Figure 1, the new digital ecosystem (called "WeDoWind") is based around specific industry-relevant "challenges", which are defined by "challenge providers" within a topical "space" and made available to participants of the ecosystem via the digital platform. The data required in order to solve a particular "challenge" are provided by the "challenge providers" under the confidentiality conditions they specify. This can include only allowing specific people to access their space, requiring them to sign agreements or preparing the data so that it is anonymous or normalised. A "challenge" is defined as a fixed problem with a motivation, goal, expected outcome and deadline. Examples of existing "challenges" being run in the ecosystem include:

- Gearbox challenge: *Participants should make use of the provided Supervisory and Data Acquisition (SCADA) data in order to train, test and validate methods that will provide clear indicators of an upcoming gearbox related fault, as well as/or a horizon-based probability of the event occurring;*
- Metadata challenge: *Propose standard metadata schemes and related semantics for sharing data in the wind energy sector in three separate steps: (1) Summarise and evaluate all existing initiatives; (2) Identify the gaps; (3) Suggest solutions for filling the gaps;*
- Brazil challenge: *Define the main problems needing solutions for implementing offshore wind energy in Brazil;*
- Diversity challenge: *Document existing resources for Diversity, Equity and Inclusion that might be useful for the wind energy community, such as guidelines, toolboxes, techniques, workshops, etc.*

If a "challenge provider" wants a more specific solution with a defined evaluation criteria and prize money, they can define a "contest" instead. If they want to discuss a more general topic, such as general experience or ideas related to a certain area, they can post a "request". This system allows participants to contribute ideas, code, data, videos and discussion topics at any time and in any form they want or can, from all over the world. They can contribute to group discussions in workshops that are run and moderated inside the space, or they can contribute digitally via discussion forums and digital whiteboards. All these communications are tagged and documented within the space. Regular emails are sent to update the participants on activities. It allows the knowledge and ideas related to a specific "challenge" to be documented and used as a knowledge base for future "challenges" on similar and overlapping topics.

The digital ecosystem **incentivises data sharing** by focusing on the needs of the people in the wind energy sector. It provides incentives both to the "challenge providers", who receive solutions to their challenges in exchange for contributing data to the platform, and to the "solution providers", who receive data in exchange for contributing ideas and

solutions to the challenges. In addition to this, the "challenge providers" get access to people and their skills for recruiting or for student projects. The "solutions providers" have the opportunity to apply and showcase their work applied to measurement data. All parties benefit from sharing ideas and innovating together. The ecosystem provides equal access to all people with an email address and internet connection, regardless of their background, education, nationality, experience, gender, and more. Furthermore, the ecosystem offers the potential to publish, develop and co-create fair methods for the evaluation of solutions. Although the community is free to develop and apply these methods themselves, the ecosystem operators (OST—Eastern Switzerland University of Applied Sciences) also intend to contribute to this in the future. This could be by introducing functionalities on the platform such as comparison-based ranking schemes for ordering posts and documents [23]. Such methods reduce the bias introduced by more popular forum ranking methods such as "star ratings" and "thumbs up-down ratings".

The method contributes to **making wind energy data FAIR** by including relevant requirements on the data that is used in the ecosystem. Part of this involves **providing a central location for data and knowledge** related to a certain topic within the sector. This allows knowledge, data, code and solutions to be centralised around a particular industry challenge and fosters collaboration and sharing. The ecosystem has the potential to **include solutions and code for data filtering and standard analysis tasks** and to **allow data standards and data structure translation solutions to be published and shared** in the future. In order to achieve this, specific "challenges" could be defined for certain applications. In the future, industry-agreed common data and metadata standards will be incorporated into the ecosystem, as well as open data analytics such as the National Renewable Energy Laboratory (NREL)'s OpenOA project (https://github.com/NREL/OpenOA; accessed on 25 July 2022).

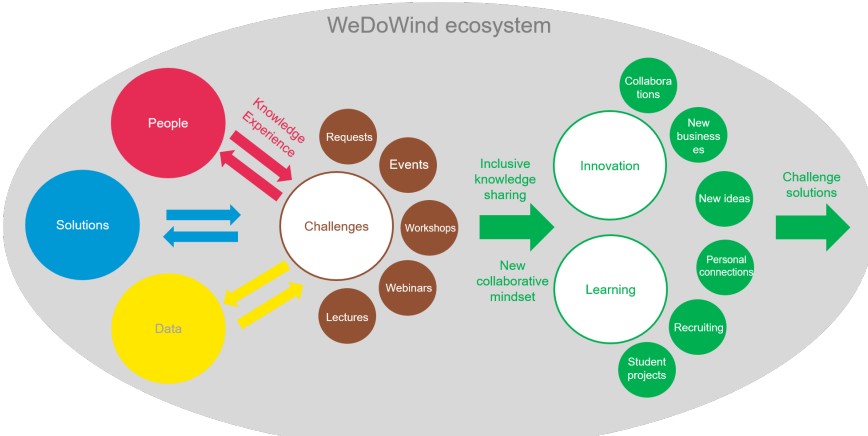

**Figure 1.** Schematic representation of the new collaboration method applied in this work, the digital ecosystem.

## 3. The Case Study

In order to test the new collaboration method, a case study was applied. This involved publishing a "challenge" in the digital ecosystem and then moderating and coordinating a co-innovation process. It resulted in a total of six new solutions. In this section, the case study is introduced, the co-innovation process is described, a literature review based on the "challenge" topic is presented, the individual solutions are described and then the results are compared and evaluated.

### 3.1. The Challenge

In this case study, a "challenge" provided by the company EDP was posted on the digital platform. The contents of the challenge are shown in italics below:

*In this challenge, we ask you to test your predictive brains and develop a global solution for this problem, focusing on the capability of detecting early-stage failures and, consequently,*

*reducing maintenance costs. The objective is to identify the failures in five of the major Wind Turbine components and advise an intervention to the wind farm operators in order to reduce corrective maintenance costs. The components to be monitored will be the gearbox, the generator, the generator bearing, the transformer and the hydraulic group. We provide two years of SCADA records from five wind turbines and data from the meteorological mast to create, train, validate, and test your models. This challenge is open until 30 September 2021! We don't plan to have a strict evaluation of the solutions submitted. Instead, our intention is to promote an open-challenge targeting a dynamic engagement with the community where new "out-of-the-box" ideas can emerge.*

The training period was defined as 1 January 2016 to 31 August 2017 and the test period from 1 September 2017 to 31 December 2017. As well as the raw SCADA data, EDP also provided nearby met mast data, a list of SCADA signal names, the data sheet of the wind turbine type and the manufacturer's power curve. A list of annotated failures and the SCADA logs were provided for the test data period. In this work, it was decided to focus on the wind turbine with the most number of annotated failures—wind turbine WT07. A summary of the annotated failures for WT07 provided by EDP for the test period for each wind turbine component is shown in Table 1. The measured wind rose, wind speed frequency distribution and power curve (without filtering) from WT07 are shown in Figure 2. The monthly averages of measurement data for WT07 in 2016, including the averages of availability, wind speed, wind speed during turbine uptime and turbine downtime, as well as box plots of power, wind speed and temperature are shown in Figure 3. The low availability during, before and after August indicates substantial downtime due to a repair. This is probably due to the annotated failure documented in the transformer in July and August 2016.

**Table 1.** Annotated failures provided by EDP for WT07.

| Component | Alarm Dates |
| --- | --- |
| Gearbox | None |
| Generator | 21 August 2017 |
| Generator Bearings | 30 April 2016 and 20 August 2017 |
| Transformer | 10 July 2016 and 23 August 2016 |
| Hydraulic Group | 17 June 2017 |

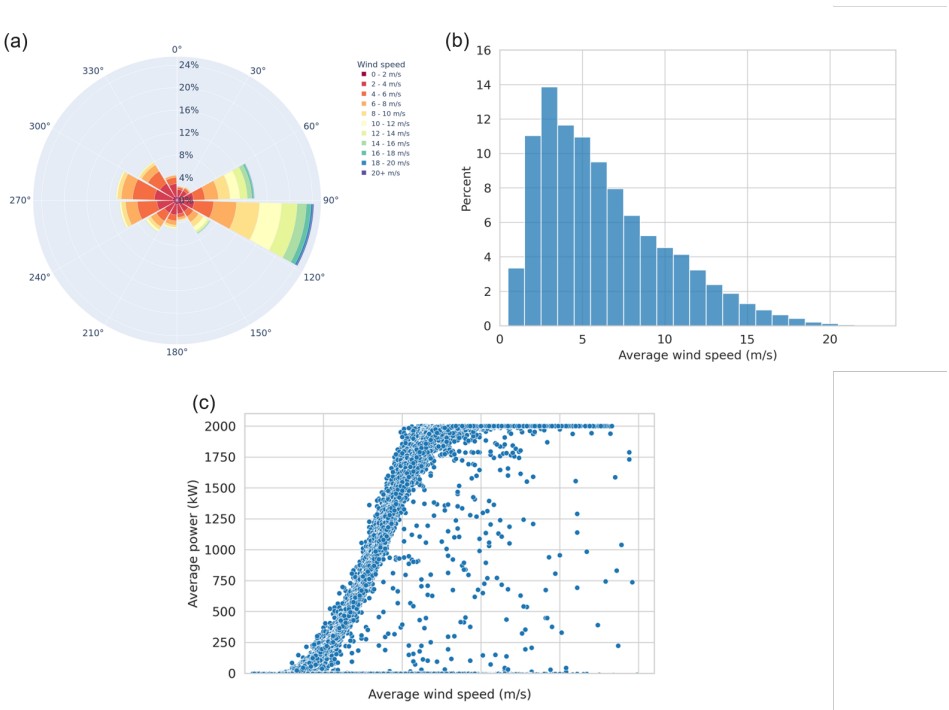

**Figure 2.** Measurement data for WT07: (**a**) wind rose; (**b**) wind speed frequency distribution; (**c**) power curve (without filtering).

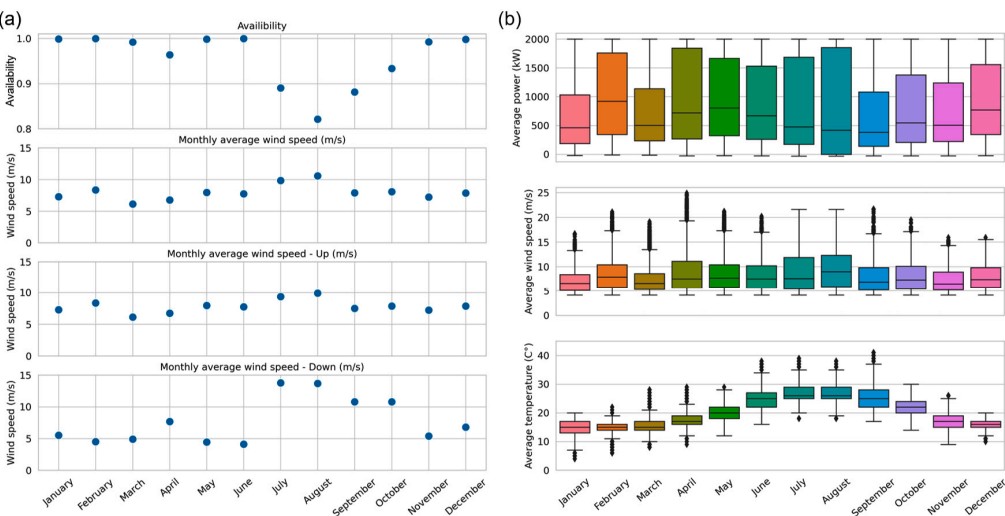

**Figure 3.** Monthly averages of measurement data for WT07 in 2016: (**a**) averages of availability, wind speed, wind speed during turbine uptime and wind speed during turbine downtime; (**b**) box plots of power, wind speed and temperature, where the centre lines show the median, the edges of the boxes the 25th percentile, the outer bars the 95th percentile and the points the outliers beyond this.

### 3.2. The Co-Innovation Process

As part of this case study, the following activities were carried out by the ecosystem operators in order to encourage a co-innovation process:

- A dedicated space called "EDP Challenges" was created on the digital platform together with EDP. The challenge description, including direct links to download the data, was developed together with EDP and posted inside this space;
- A public "call for participants" website was created with a direct link to the registration form. This was shared within the wind energy community using social media;
- A process for allowing EDP to decide who may participate or not was set up. This process was not meant to reduce accessibility to the challenge, but instead to ensure that applicants were real people interested in the challenge and not robots, bots or imposters;
- A "Getting Started Guide" to using the digital platform was created and explainer videos were recorded in order to help users interact on the platform;
- A series of online workshops were organised for the participants—a launch workshop, interim workshops every month and then a final workshop. These involved brainstorming sessions in small groups as well as question and answer sessions with EDP. The sessions were documented on a digital whiteboard and recordings were posted in the digital space;
- Regular email updates were sent with specific questions and actions to encourage interaction. This included requests to summarise and comment on different possible methods, as well as discussions of evaluation methods;
- The space was regularly checked, cleaned and coordinated by the ecosystem operators to ensure that the information was up-to-date and understandable;
- Regular updates were communicated on social media during the challenge.
- A downloadable docker was made available to allow beginners easy access to the data and code. This was integrated into a smaller "sub-challenge" run at the Eastern Switzerland University of Applied Sciences.

### 3.3. Existing Wind Turbine Fault Detection Methods

Before the solutions submitted to this challenge are introduced, existing methods for wind turbine fault detection as well as for model evaluation are reviewed here.

### 3.3.1. Wind Turbine Fault Detection Methods

In general, condition monitoring of wind turbines is an integral part of the operation and maintenance (O&M) of the asset. Avoiding component failure can save the asset owner large amounts of money. For example, an analysis of over 300 offshore wind turbines and found that failure rate per offshore turbine per year is about 10, with around 80% requiring minor repairs (<€1000), 17.5% major repairs (€1000–€10,000) and 2.5% major replacements (>€10,000) [24]. In addition to this, the same study identified the pitch/hydraulic, generator and other subsystems as contributing the most to failure rates. Generators and converters tend to have a higher level of failure rates in offshore wind turbines than onshore ones.

Maintenance can be reactive, preventive or predictive [25]: reactive maintenance involves waiting until a component fails before replacing it, and does not involve any active monitoring, preventative maintenance involves scheduled replacements, and predictive maintenance involves monitoring components and predicting failures before they happen. Components can be monitored using the standard SCADA data produced by a wind turbines [26], as well as using specialised higher frequency measurement equipment such as drive train vibration sensors [27], oil debris monitoring and rotor blade pressure sensors [28].

The utilisation of operational SCADA data for condition monitoring has attracted considerable research interest since it provides insights without the need for additional equipment. For this, Machine Learning (ML) can be used to build an inductive model that learns from a limited amount of data without specialist intervention. In order to do this, an underlying set of structures or patterns are found, which help understand relationships in data that cannot be otherwise detected. So-called 'supervised learning' predicts an output variable using labelled input data, whereas 'unsupervised learning' infers relationships from data without labelled inputs. Supervised learning models can be categorised into regression and classifiers, where regression models predict a numeric variable and classifiers predict a categorical variable. In addition to the two categories, a semi-supervised learning approach can be used when data points are partially labelled, for example by training the model on the normal data and classifying future observations as anomalies when they deviate from the normal. Examples of this category include the use of residuals from the modelled normal data on control charts to determine abnormality [29–32].

A wide range of ML methods has proven to be able to detect developing malfunctions at an early stage, often months before they resulted in costly component failures (see, e.g., [20,33–35]. For a comprehensive review, refer to [36]. SCADA data-based condition monitoring, therefore, represents a cost-efficient and effective complement to state-of-the-art condition monitoring solutions. Its primary task is to classify the state of a turbine or one of its components as either healthy or faulty. However, most available SCADA data represent predominantly healthy operation with no or only comparatively few instances of faulty conditions. In such a setting, semi-supervised anomaly detection, often called normal behaviour modelling, has proven to be useful e.g., [37]. Normal behaviour models (NBMs) are trained on healthy data to represent the class corresponding to the normal state. Subsequently, deviations between model output and the measured sensor values can be processed and evaluated to identify anomalies. For wind turbines, performance and temperature monitoring can be distinguished. The former aims to detect abnormal deviations from the turbine's usual power output, whereas the latter aims to detect deviations from the healthy thermal equilibrium conditions. Temperature monitoring is better suited for detecting malfunctions in the components along the drive train, which account for the majority of turbine downtime [38]. Ref. [33] was among the first to apply the approach in the wind domain and prove its feasibility. Many publications with successful early detection of malfunctions followed, e.g., [34,35,39–42]. However, no particular method has yet been established as being optimal, due to the difficulty of comparing and quantifying the performance of different methods.

### 3.3.2. Model Evaluation Methods

Evaluation of time series anomaly detection (TSAD) algorithms, as required for fault detection problems, is a challenging task. One reason is that classical anomaly detection metrics were originally designed for point-based anomalies, whereas, in TSAD, we often encounter range-based anomalies that are present for a certain period of time [43]. Another reason is that algorithm performance is often highly sensitive to the required choice of alarm threshold [44]. Lastly, false and missed alarms can have very different implications, depending on the domain, and are therefore difficult to compare across applications.

Recent literature reported various evaluations on wind turbine fault detection. Most of those evaluations are based on the distance or the difference between actual output ($Y$) and predicted output ($\hat{Y}$). When SCADA data are used as input in the model, the evaluation is commonly point-wise. For regression-based normal behaviour models, the most common measures include mean absolute error (MAE), mean absolute percentage error (MAPE) [45], and root mean squared error (RMSE) [46]. Classification models are typically evaluated using accuracy, sensitivity, specificity and F1-measure [47–49]. All of the aforementioned measures are evaluating the methods without taking into account how it will cost or benefit the industry. Thus, it does not provide a direct estimate of potential savings when a detection is made.

In order to transfer prediction performance to cost savings for the asset owner, the costs and savings due to the use of a particular model compared to not using it have to be estimated. As this step is very specific to the asset owner, there is no agreed-upon method for doing this in the literature. On the EDP Open Data platform [50], the following method is used for fault detection of subsystems within wind farms:

*Step 1: The predicted faults for each wind turbine and subsystem (e.g., gearbox, generator, etc.) are classified as follows:*

- True positives (TP): a failure of the correct wind turbine and subsystem is correctly predicted between two and 60 days before the actual failure;
- False negatives (FN): an actual failure is not detected between two and 60 days in advance;
- False positives (FP): a failure is predicted that does not actually occur in the next two to 60 days.

*Step 2: Each detection type is converted into costs as follows:*

- True positives (TP): translated into savings, $TP_s$, which are the difference between replacement costs, $Cost_{rpl}$, and repair costs, $Cost_{rpr}$;
- False negatives (FN): translated into costs, $FN_c$, due to replacements, $Cost_{rpl}$;
- False positives (FP): translated into costs, $FN_c$, due to inspections, $Costs_{insp}$.

The replacement, repair and inspection costs assumed by EDP on their Open Data Platform are summarised in Table 2.

**Table 2.** Summary of costs assumed for the EDP evaluation method.

| Component | $Cost_{rpl}$ (€) (Replacement Costs) | $Cost_{rpr}$ (€) (Repair Costs) | $Costs_{insp}$ (€) (Inspection Costs) |
|---|---|---|---|
| Gearbox | 100,000 | 20,000 | 5000 |
| Generator | 60,000 | 15,000 | 5000 |
| Generator Bearings | 30,000 | 12,500 | 4500 |
| Transformer | 50,000 | 3500 | 1500 |
| Hydraulic Group | 20,000 | 3000 | 2000 |

*Step 3: The total prediction savings are calculated:*
The costs or savings for each detection type are then summed as follows:

$$TP_s = \sum_{i=n_{TP}} \left( Cost_{rpl} - \left( Cost_{rpr} + (Cost_{rpl} - Cost_{rpr})(1 - \Delta t / 60) \right) \right) \tag{1}$$

$$FN_c = n_{FN} \times Cost_{rpl} \qquad (2)$$

$$FP_c = n_{FP} \times Cost_{insp} \qquad (3)$$

where $n_{FN}$ = total number of false negatives and $n_{FP}$ = total number of false positives. The Total Prediction Savings, TPS, are then given as follows:

$$TPS = TP_s - FN_c - FN_c \qquad (4)$$

This number represents the potential of a given prediction tool for reducing (preventive and corrective) maintenance costs. It is used on the EDP Open Data platform in order to compare and rank submitted solutions, and is used in the present work as well.

### 3.4. Description of the Submitted Solutions

In this section, the six different solutions submitted as a response to the challenge are described. A summary of the solutions is given at the end of the section, together with a discussion and comparison of the data pre-processing methods.

### 3.4.1. Normal Behaviour Models (NBM)

As mentioned in the previous section, normal behaviour models (NBMs) learn from historical data and can be used to infer what should be the turbine's normal operating condition. If the actual measured values from that same sensor deviate too much from the NBM's prediction, it means that the turbine is operating in an abnormal condition, and therefore an alarm is raised by the algorithm. As a given component starts to degrade and a failure mode starts building up in the turbine, measured values of temperatures and other sensors may start increasing in a way that is not perceptible to the naked eye but can be captured by the mathematical analysis of this algorithm.

While the NBMs are regression-based models and make predictions for the turbine sensors, these are estimations of how the turbine should be behaving at a distinct time period, given the other available measurements. These predictions are not forecasts for the following days or weeks; they are "hindcasts" to check if recent turbine operation fits inside the normal operation threshold or not. Since this approach is aimed at monitoring a large fleet, comprising hundreds of wind turbines each with dozens of sensors, the NBMs employ a linear regression to model the relationship between the different sensors. This way, both the training process and the daily predictions can be done very quickly. Previous studies [20] have showed that linear regression, although somewhat simple, is an acceptable choice. An initial version of this algorithm [51] used ensemble models, but an internal study concluded that the gain in prediction accuracy was small when compared to the increase in computational cost.

The selection of inputs to predict any of the turbine sensors is done manually, from expert knowledge because an automatic selection algorithm based on correlation between the candidate inputs and the target sensor may be misleading in some cases. For example, when predicting the temperature of generator winding 1, such algorithm would probably choose the temperature of winding 2 or 3 as the highest-correlated candidate input. However, it would be an input that adds no information to the system because any kind of generator failure or degradation that leads to overheating would cause this effect on the three windings. Therefore, if winding 2 increases in temperature and it was used as input, it would lead to a higher predicted temperature for winding 1, and the failure would remain undetected.

The objective of the training process is to minimise the error between the measured values and the model predictions (for each sensor). After training, in the prediction stage, this error is averaged for each day, to condense results and reduce uncertainty and the effect of possible outliers. A sensor is classified as presenting an anomaly if the normalised daily error (using the daily error's mean and standard deviation calculated from the

training process) exceeds a ±3 standard deviation range, meaning it is probably not part of the expected error distribution. To avoid false positives and reduce the algorithm sensitivity, an alarm is only raised to the user if three or more of the past seven days are classified as anomalous. Figure 4 shows an example where an inverter cooling fan malfunction was detected by the algorithm. The prediction error increases when the fan starts malfunctioning, and alarms are sent to the operator everyday, until the problem is corrected and the prediction error is reduced.

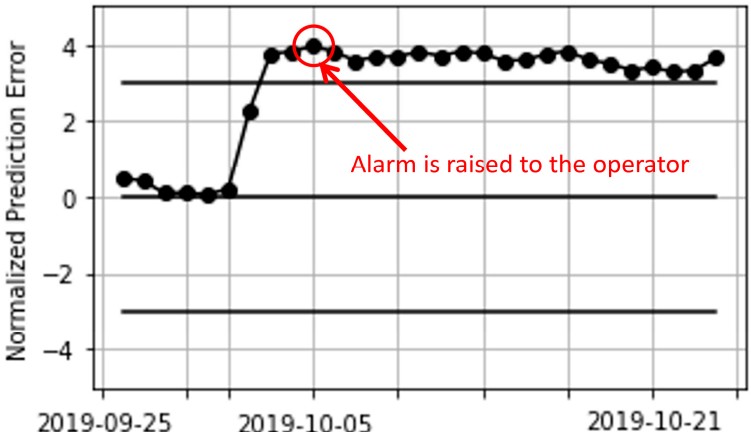

**Figure 4.** Calculated prediction error for an inverter fan malfunction for the NBM model. The algorithm sends daily alarms to the operator until the problem is corrected and the error returns to the acceptable range.

### 3.4.2. Combined Local Minimum Spanning Tree and Cumulative Sum of Multivariate Time Series Data (LoMST-CUSUM)

Cumulative sum (CUSUM) is a memory-type control chart that works by accumulating consecutive sample points over time to monitor changes in process. It is particularly known to effectively detect a small shift in the process that memoryless methods would normally fail to detect. Due to its ability to accumulate effectively small-magnitude early symptoms over time for symptom tracking, CUSUM principles are adopted in this approach. CUSUM-based approaches have been used in wind turbine monitoring in combination with ML plots [30,31]. This approach also employs a chart that works like CUSUM control-chart, as a mechanism to raise alarms as a warning that failures are potentially going to happen.

The classic CUSUM-chart uses samples measurement to establish the monitoring plot. Most of the current approaches that employ CUSUM use a normal model residuals to establish the plot. In this approach, the chart takes anomaly scores that are produced by an unsupervised algorithm called Local Minimum Spanning Tree (LoMST) [52]. In order to implement this CUSUM-inspired mechanism, three parameters need to be defined to establish the chart. First, the offset that sets the boundary between the normal and anomaly points; only those above the offset should be plotted on the chart. Second, the accumulation windows that set the maximum time in which two consecutive anomaly points above the offset will be considered to come from the same cluster of alarms. When two anomaly points are far apart beyond the predefined time windows, the cumulative score will be reset to zero and a new cluster of accumulation will begin again. Third, the threshold that defines the minimum cumulative scores to be considered as alarms. Any cumulative scores that do not reach the threshold will not be considered as an alarm. In brief, this threshold acts as a boundary that raises alarms to the possible future failure.

The LoMST algorithm works in three stages as described in [52]. First, it establishes a so-called Minimum Spanning Tree (MST) using all data points. Second, it isolates the cluster anomalies by removing the links of the global MST one by one. Third, it repeats the second step to identify point-wise anomalies. At the end, an outlier score is assigned to each of the data points, indicating the anomaly level of the point. Because LoMST is

an unsupervised learning method, it uses the structure or pattern of the data instead of data labels to identify any anomalies. This is an advantage because labelling can be a very challenging task. Additional advantage is that it takes multivariate input, and the output is a univariate anomaly score. This univariate anomaly score simplifies how the chart should be designed.

The implementation of LoMST-CUSUM requires the three aforementioned parameters called the offset, the accumulation window, and the threshold. These parameters are defined based on the training data by striking the balance between maximising the failures detection and minimizing the false positives. Figure 5 illustrates how an alarm is raised using this approach. Latiffianti et al. [53] presents a detailed account of LoMST-CUSUM approach and its implementation for wind turbine gearbox failure detection.

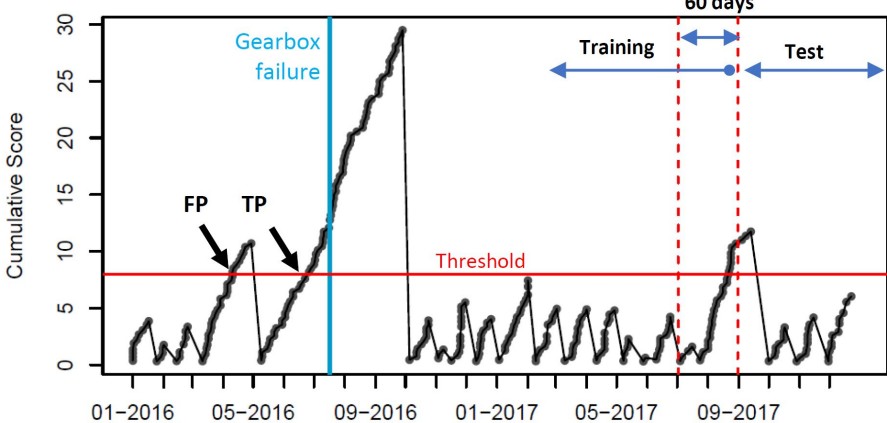

**Figure 5.** Illustration of how the combined LoMST and cumulative sum of anomaly score (LoMST-CUSUM) is used to establish failures detection. In this example, the blue line indicates the time at which a gearbox failure happened. The parameters are defined based on the training data in five turbines.

### 3.4.3. Combined Ward Hierarchical Clustering and Novelty Detection with Local Outlier Factor (WHC-LOF)

This solution combines two methods to detect the turbine failure by comparing the parameters of a group of wind turbines based on the SCADA data (e.g., nacelle temperature of five wind turbines). The first method is the Ward Hierarchical Clustering [54], where the *AgglomerativeClustering* algorithm setup with ward mode from the Python *sklearn* package was used to separate normal and anomalous conditions in twelve clusters. The 'normal' condition is considered when the parameters of the wind turbines are similar (e.g., nacelle temperature is the same for all turbines). When the parameter of one wind turbine is significantly different from the other wind turbines, this cluster is classified as an anomalous condition. The number of neighbours is a parameter that can be tuned in the algorithm according to the similarity of each cluster, where the number twenty was used for this case. Thus, the training data are filtered using only the normal condition data, and the clusters identified as anomalous conditions are removed.

The second method is the Novelty Detection with Local Outlier Factor (LOF) [55], which is used to detect the outliers associated with the failures of the wind turbine. The training data, pre-processed by the Ward algorithm, is used to training the *LocalOutlierFactor* algorithm from the Python *sklearn* package. Thus, any new data from the test data that do not match with the 'normal' condition is detected by the LOF algorithm as an outlier. The novelty detection mode is configured in LOF instead of the outlier detection mode because the outlier detection mode can only identify outliers found in training data, whereas the novelty detection mode can detect unknown outliers, which is any data that are not considered 'normal' in the training data. Figure 6 shows an example of a turbine failure detected using the WHC-LOF method.

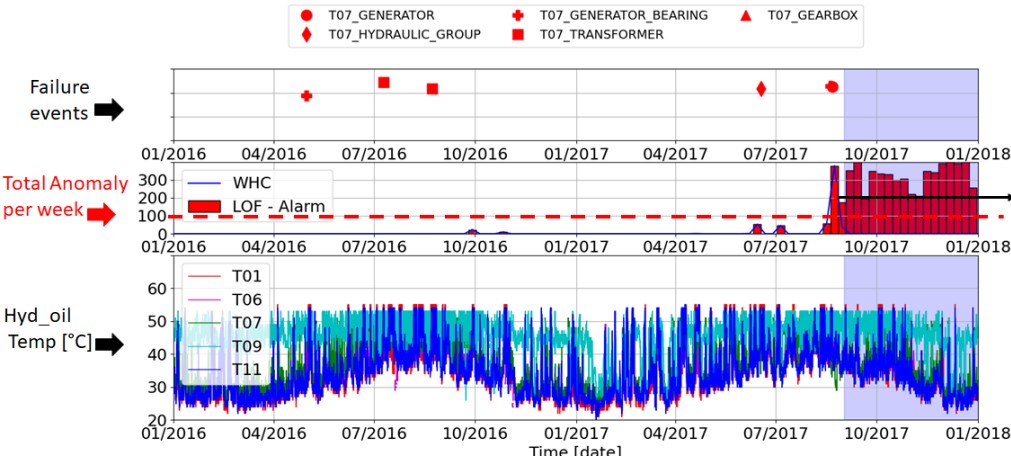

**Figure 6.** Time series of hydraulic oil temperature of the five turbines and the total anomaly per week detected by the WHC-LOF method (red bar). The red dashed line is the threshold alarm when the cumulative anomaly event is above 100/week. The black arrow indicates the expected failure in the next weeks.

3.4.4. Normal Behaviour Model with Lagged Inputs (NBM-LI)

This solution uses a random forest regression model to predict normal turbine behaviour by incorporating information from previous times. When the predicted turbine behaviour deviates significantly from the observed behaviour, an alarm is raised. The Python *xgboost* random forest library was used, specifying 50 trees and a maximum depth of two, and otherwise using the default parameters.

New signals were added to the data set, lagging the original SCADA data channels by periods of 10, 20, 30, 40, 50, and 60 min. Furthermore, signals were added corresponding to the ratio between the original SCADA signals and the corresponding values occurring in these past times. Including these additional features generally reduced the error of the prediction model, as shown in Figure 7, which plots the L-2 norm of the errors associated with prediction of the generator slip ring temperature as a function of the number of trees used by the random forest model (80% of the training data was used to train the regression model and the remaining 20% was used to compute the error). While using all the available SCADA signals achieved a lower error, it was decided to only use the generator speed and power produced in the final model to avoid overfitting.

The testing data set was used to predict normal behaviour of the generator slip ring temperature for each 10 min interval. The recorded slip ring temperature was then used to compute the absolute error of each prediction. Errors larger than 15 times the standard deviation of the errors associated with these predictions were flagged as anomaly events. The random forest was initialised with a single random number generator seed, and could be improved by considering the aggregate of several random number generator seeds.

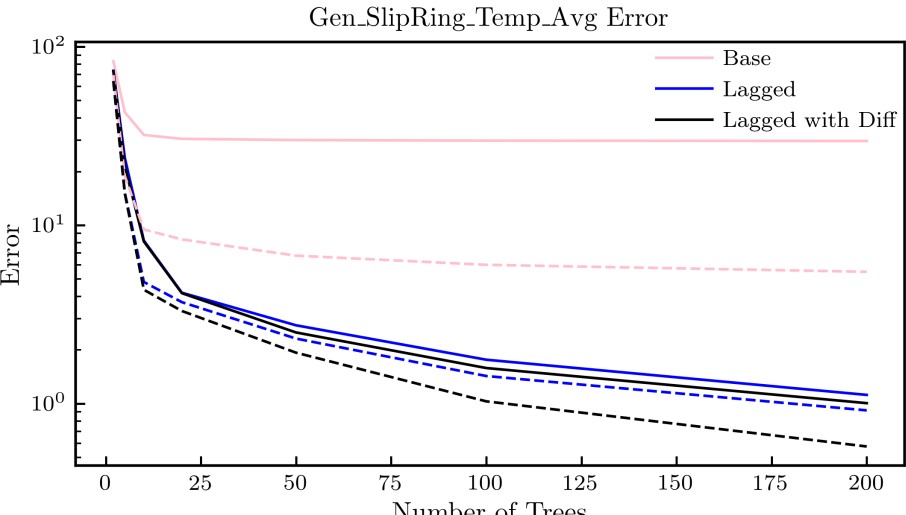

**Figure 7.** L-2 norm of the error of predicted generator slip ring temperature plotted as a function of trees in the random forest model (NBM-LI). The solid lines show the results associated with only using the generator speed and power produced. The dashed lines show the results associated with all SCADA signals, excluding the generator slip ring temperature. The different colors represent results associated with the baseline data set (without feature engineering), lagging the data set, and examining the ratio of current to previous signal values in addition to lagging the data.

### 3.4.5. Canonical Correlation Analysis (CCA)

Canonical correlation analysis (CCA) focuses on maximising the correlation between two sets of variables for fault detection [56]. In detail, training and testing samples are first collected from input and output measurements, which are then standardised as input and output matrices. The basic idea of CCA is to seek two weighting matrices to maximise the correlation between input and output matrices, in which the singular value decomposition is hence leveraged to achieve this. Finally, the residual vector is constructed by the weighting matrices to obtain the squared prediction error (SPE) statistics. This will reflect the trend of the system operation.

For the threshold design, the kernel density estimation (KDE) [57] is used to bound the residual vectors in CCA. In principle, it is a non-parametric technique to estimate the characteristics of a certain probabilistic distribution. As KDE is able to solve the problem of non-Gaussian assumptions, it has been widely used in fault detection techniques [58]. In CCA, the threshold is determined based on the underlying probability density function derived from the statistics. In the fault monitoring of WT07, the way of selecting input variables is the same as for NBM, and then the CCA model is trained with the healthy samples. As shown in Figure 8, the SPE statistics will exceed the threshold, which indicates the faults of the wind turbine.

### 3.4.6. Kernel Change-Point Detection (KCPD)

This solution detects change points (CPs) in single SCADA signals before any model is trained. Therefore, it enables the exclusion of periods contaminated by previously present faults or malfunctions. Each analysed signal is prepared by removal of non-operational periods, a normalisation with respect to operational state as well as ambient conditions, and a final re-sampling with reduced temporal resolution. Then, a kernel change-point detection algorithm is applied in order to screen the prepared signals and flag changes induced by irregular variations of the underlying physical system. The methodology is described in greater detail in [59]. Note that the method works offline and is therefore not suitable to predict failures, but to detect them in existing training data sets. Therefore, this solution is evaluated separately from the other online methods.

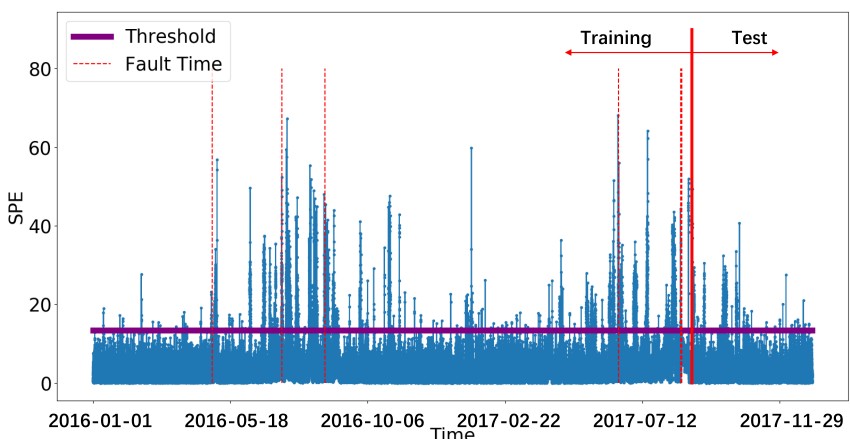

**Figure 8.** An example of the fault monitoring results of CCA, where the residual vector is represented by the SPE statistics and the threshold is computed by KDE.

Application of the algorithm to the data from WT07 with the settings suggested in [59] resulted in the detection of two CPs (compare Figure 9). The first one coincides with the reported damage of generator bearings and is detected in the temperature measurement of generator bearing 2, therefore providing additional information as to which of the two bearings was presumably affected. The second CP is detected in early November 2016 in one of the transformer phase temperatures which was reported to have been abnormally high in the preceding months. Visual inspection of the processed signal indeed confirms a change in the behaviour. Additionally, the SCADA log files show several hours of downtime and local access to the turbine on the day the change point was flagged. However, the operator has not reported any relevant maintenance activity in close temporal proximity and therefore conclusions about the change-points origin remain speculative despite the suggestive evidence from the data.

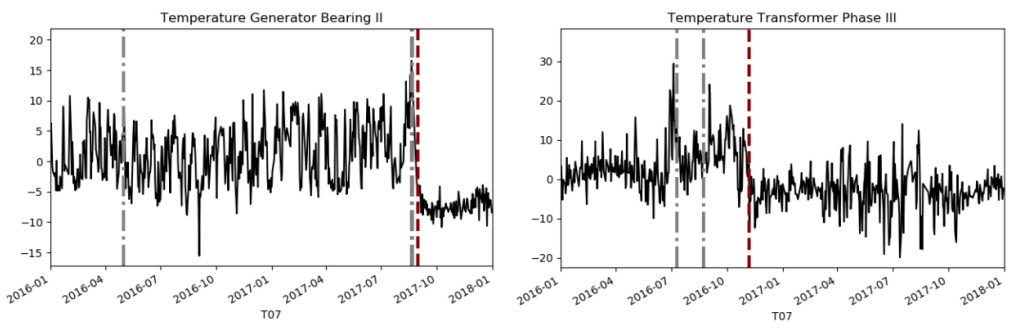

**Figure 9.** Results of the KCPD algorithm for the generator bearing (**left**) and the transformer (**right**) with processed measurement over time (black), annotated malfunctions of the respective component (grey dashed), and detected change points (red dashed).

3.4.7. Summary of Solutions

In Table 3, a summary of the solutions in terms of the solution providers, the method type, the detection type, the previous application to wind turbines and if it is used for the comparison in this paper.

**Table 3.** Summary of the solutions examined in this work.

| Solution | NBM | LoMST-CUSUM | WHC-LOF | NBM-LI | CCA | KCPD |
|---|---|---|---|---|---|---|
| **Contributer** | Voltalia, France | TAMU, USA | Fed. Inst. Santa Catarina, Brazil | Univ. Colorado, USA | TU Delft, Netherlands | TU Berlin, Germany |
| **Type ("S" = Supervised, "U" = Unsupervised, "SS" = Semi-supervised)** | S | SS | S | S | U | U |
| **Real time?** | Yes | No | Yes | Yes | Yes | No |
| **Type of detection ("PW" = Point-wise, "CB" = Chart-based)** | PW | CB | PW | CB | CB | CB |
| **Previous application to wind turbines?** | Yes [51] | No | No | No | Yes [60] | Yes [59] |
| **Used in comparison?** | Yes | Yes | Yes | Yes | Yes | No |

Additionally, it is important to discuss and compare the data pre-processing methods because data quality is a major concern in SCADA data analysis [61]. Researchers have developed various approaches ranging from manual data screening [33] via automated threshold checks [20] to advanced statistical filtering methods [34]. The discussion between the solution providers has revealed that results are indeed often sensitive to pre-processing settings, which is why we want to give a concise overview on the approaches taken. Mandatory data quality checks, such as identification of missing values, constant values or parameter range checks were mostly conducted manually by domain experts. Other choices for automated pre-processing were filtering out non-operational periods, reduction of temporal resolution, unsupervised clustering methods and iteratively excluding data points with poor training performance from the training set. Table 4 gives an overview of the pre-processing methods applied within each solution in the present work. In general, we encourage reporting pre-processing in detail, due to their importance for reproducing reported results.

**Table 4.** Overview of data pre-processing approaches by solution.

| Solution | NBM | LoMST-CUSUM | WHC-LOF | NBM-LI | CCA | KCPD |
|---|---|---|---|---|---|---|
| **Filtering** | Iterative during training | Manual/Domain expert | Ward Cluster Algorithm | Manual/Domain expert | Manual/Domain expert | Non-operational based on power |
| **Time resolution** | 10 min | 1 h | 10 min | 10 min | 10 min | 24 h |

*3.5. Evaluation of Solutions*

The solutions were first evaluated using EDP's method described in Section 3.3.2 [50]. EDP's own model is included in the analysis as well as the five models NBM-LI, NBM, LoMST-CUSUM, CCA and WHC-LOF. No information about EDP's own model is known to the authors.

The results of the predictions for the wind turbine WT07 for each model applied to the training data and the test data are shown in Figure 10. The coloured dots mark the dates of the predicted failure of each component considered for each model, and the area shaded blue on the right marks the test period. The red circles refer to the annotated failures provided (labelled "SCADA"). The one annotated failure that was identified in the test period was not known to the participants of the challenge, and therefore the test period represented a blind test. However, due to the very short test period and the corresponding lack of annotated failures during this period, both periods will be considered in this analysis.

For CCA, the detected faults in the test period were not specific to a particular component and therefore are labelled "Fault".

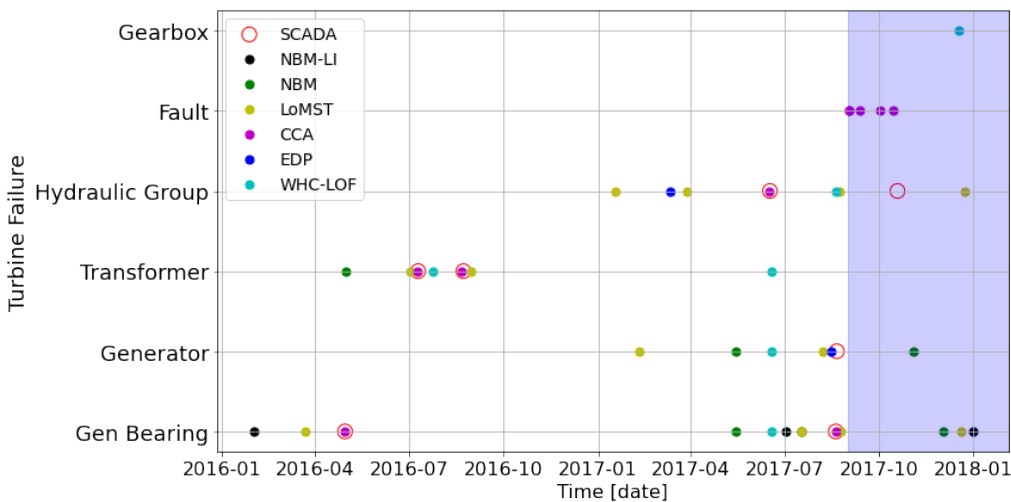

**Figure 10.** Comparison of fault predictions for each model for WT07.

The time between the predicted alarms and each annotated failure was first calculated for each component and model. Figure 11 shows this time in the upper figure (a negative value refers to "before") for each component and model using the same colours as the previous plot. The lower plot shows a frequency distribution of these times, split into bins of 30 days. The most frequent time differences are in the −30, 0 and 30 day bins, as expected.

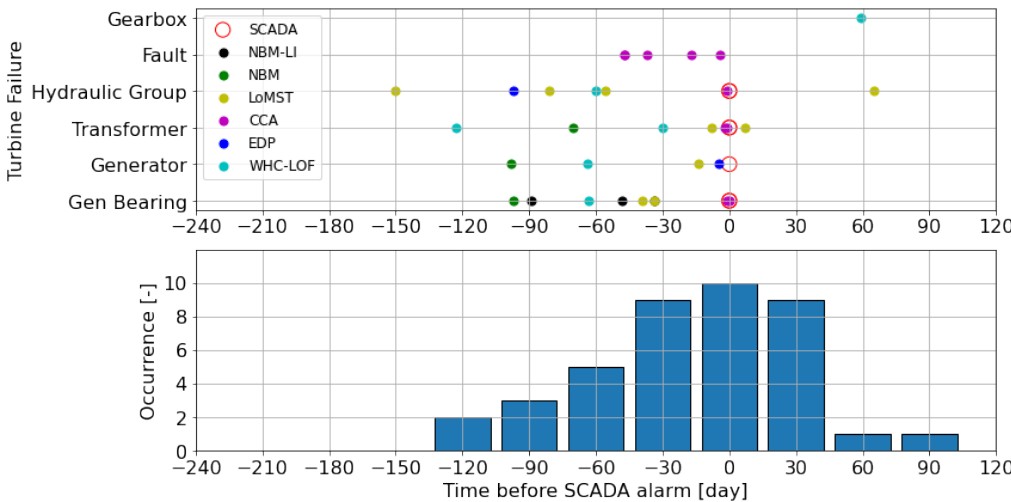

**Figure 11.** Time before annotated failure for each model and the its distribution.

Next, each predicted fault was classified as True Positive (TP), False Negative (FN) or False Positive (FP) as described in Section 3.3.2. A summary of the number of each type of fault for each model is given in Table 5, split into the training and test periods. For the CCA model in the test period because the component experiencing the alarm could not be identified, the first fault prediction within 2–60 days before an annotated failure was treated as a TP, and the savings were calculated for the damaged component (the Hydraulic Group). In addition to this, the same prediction was assigned an FP, and inspection costs were included for all the components except for the Hydraulic Group. Further faults detected after this were classified as FPs, and inspection costs were included for all the components except for the Hydraulic Group.

**Table 5.** Number of each type of predicted fault for each model for the training and test periods.

|  | NBM | | LoMST-CUSUM | | WHC-LOF | | NBM-LI | | CCA | | EDP | |
|---|---|---|---|---|---|---|---|---|---|---|---|---|
|  | Train | Test | Train | Test | Train | Test | Train | Test | Train | Test | Train | Test |
| TP | 0 | 0 | 4 | 1 | 1 | 1 | 1 | 0 | 1 | 1 | 1 | 0 |
| FN | 6 | 1 | 2 | 0 | 5 | 0 | 5 | 1 | 5 | 0 | 5 | 1 |
| FP | 3 | 2 | 7 | 3 | 2 | 2 | 2 | 1 | 0 | 4 | 1 | 0 |

The resulting savings due to TPs and the costs due to FNs and FPs as defined in Section 3.3.2 are summarised in Figure 12. For the training period, it can be seen that the FN costs dominate for all models. This is because it is assumed that FNs lead to replacement costs, which are high. For the test period, the costs and savings are generally much lower because only one annotated failure occurred, and the models predicted fewer FNs. The dominating fault type is FP, which are assumed to lead to an inspection.

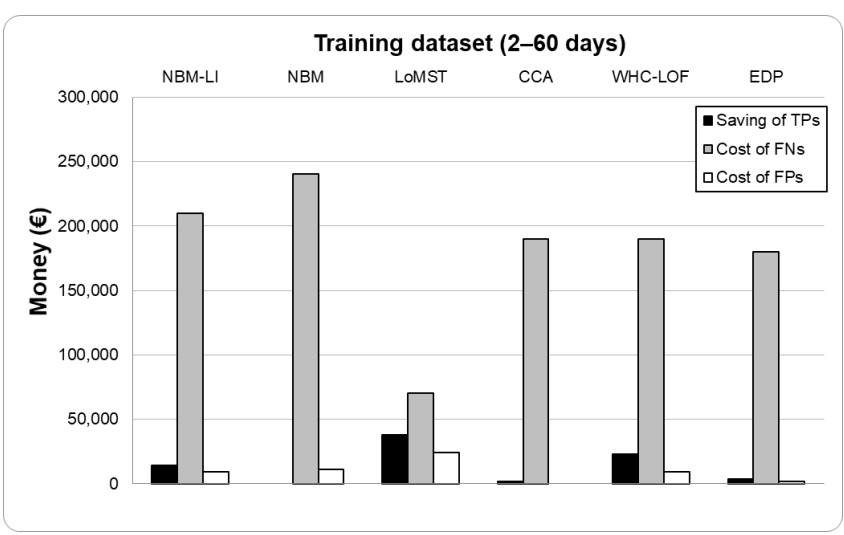

(**a**) Training period

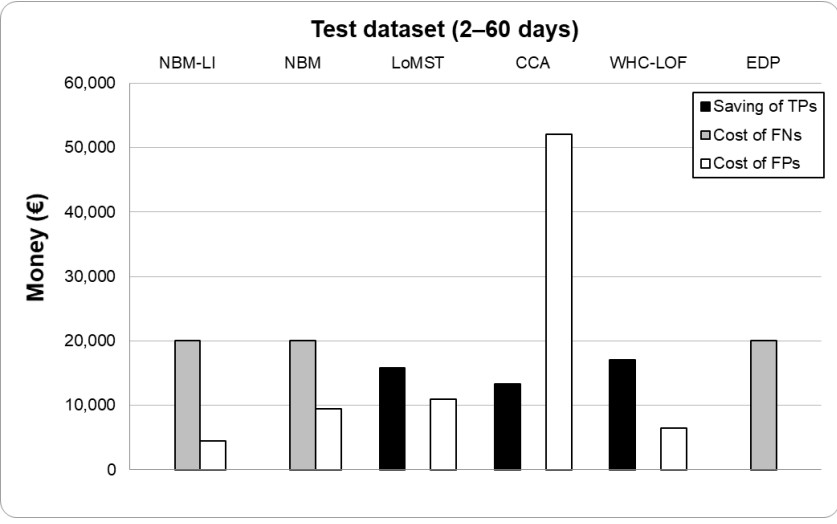

(**b**) Test period

**Figure 12.** Savings and costs due to different fault type for each model, EDP evaluation method (2–60 days).

The Total Prediction Savings (TPS) as defined in Section 3.3.2 are shown in Figure 13 for each model for the training and test periods. A positive value refers to positive savings compared to the situation if no prediction tool would be used. It is very interesting that all the models lead to losses for the training period, ranging from €50,000 to €250,000 depending on the model. This means that, even models that have been trained with historical data could perform worse than no predictions, and asset owners should not automatically assume that prediction tools lead to savings. For the training data, two of the models, LoMST-CUSUM and WHC-LOF, lead to small savings of up to €10,000 compared to no predictions. The other models, including the EDP model, all lead to losses between €20,000 and €40,000. It should be noted that, because the time period was short and only one failure occurred in this time, these results should be treated with care. Further analyses over longer periods would increase the confidence in the results.

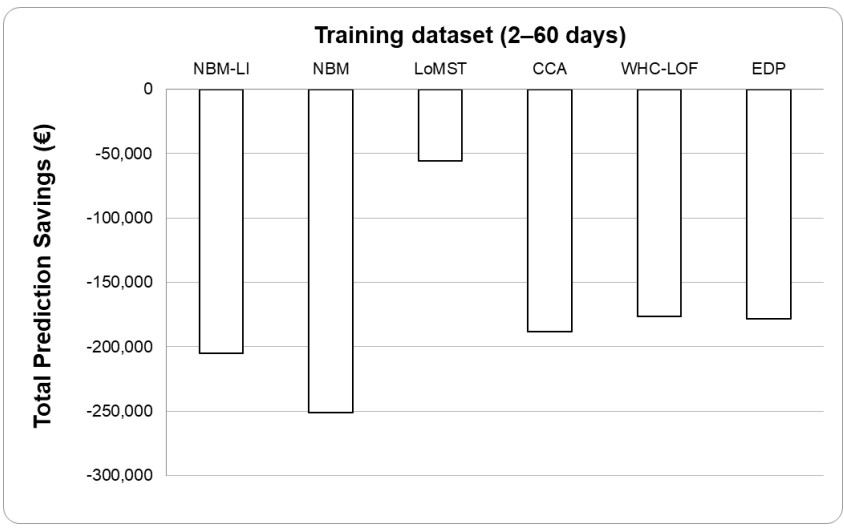

(**a**) Training period

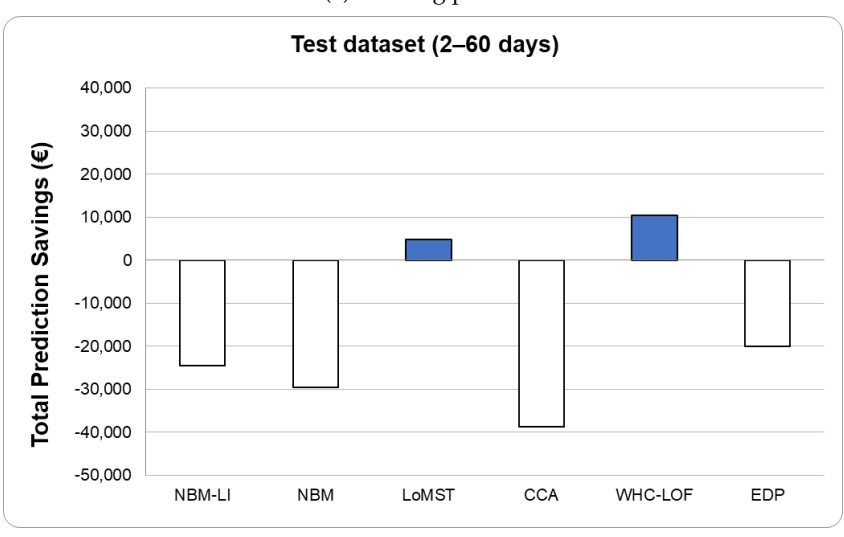

(**b**) Test period

**Figure 13.** Total Prediction Savings (TPS) for each model, EDP evaluation method (2–60 days).

In order to quantify the value of the results of this challenge to the challenge providers EDP, the differences between the TPS obtained with each model submitted for the challenge and the TPS using the EDP model were calculated. These represent the expected savings brought by a switch from the EDP model to another model, and are summarised in Tables 6 and 7. The LoMST-CUSUM model performs significantly better than the EDP model for the training period, and would have saved EDP €122,242. For the test period,

which is more important for assessing performance for unknown faults, the LoMST-CUSUM model saves EDP €24,867 and the WHC-LOF model would save them €30,500.

In addition to this, the difference between the average of all the TPS values obtained with each model and the TPS using the EDP model was calculated to be €2424 for the training period and €3781 for the test period. This represents the improvement EDP would expect using this collaboration method rather than if they had chosen one random partner from the five.

**Table 6.** Total  Prediction Savings (TPS) and improvement over EDP ($\Delta TPS$) for each model for training period.

|  | Average | NBM | LoMST-CUSUM | WHC-LOF | NBM-LI | CCA | EDP |
|---|---|---|---|---|---|---|---|
| **TPS** | −€175,826 | −€251,000 | −€56,008 | −€176,250 | −€205,000 | −€188,450 | −€178,250 |
| $\Delta TPS$ | - | −€72,750 | €122,242 | €2000 | −€26,750 | −€10,200 | - |

**Table 7.** Total Prediction Savings (TPS) and improvement over EDP ($\Delta TPS$) for each model for test period.

|  | Average | NBM | LoMST-CUSUM | WHC-LOF | NBM-LI | CCA | EDP |
|---|---|---|---|---|---|---|---|
| **TPS** | −€16,219 | −€29,500 | €4867 | €10,500 | −€24,500 | −€38,683 | −€20,000 |
| $\Delta TPS$ | - | −€9500 | €24,867 | €30,500 | −€4500 | −€18,683 | - |

## 4. Discussion of Results

The application of the EDP evaluation method has allowed the submitted solutions to the case study challenge to be compared and evaluated. However, the evaluation method includes several assumptions, some of which are investigated in this section. Following this, each method is evaluated qualitatively, further challenges of the evaluation method are discussed and, finally, the new collaboration method itself is discussed.

### 4.1. Assumptions of the EDP Evaluation Method

The following key assumptions were thought to affect the evaluation results:

1.  A predicted alarm may lead to savings if detected even earlier than 60 days before the fault. Figure 14 shows the effect of altering the definition of TP from 2–60 days to 2–90 days (including adjusting Equation (1));
2.  A predicted alarm may lead to savings if detected even later than two days before the fault. Figure 14 shows the effect of altering the definition of TP from 2–60 days to 1–90 days;
3.  It may very well be the case that not every annotated failure leads to a failure that requires complete replacement or a component. This would reduce the costs of an FN. Figure 14 shows the effect of halving the replacement costs for each component on the TPS for each model (using 2–60 days);
4.  An asset owner may decide not to inspect repeating alarms for the same components. This would reduce the number of FPs.  Figure 14 shows the effect of removing inspection costs for repeat alarms for each component on the TPS for each model (using 2–60 days).

The effect of the variations on the TPS are different depending on the model. Altering the TP period from 2–60 days to 2–90 days generally has a positive effect on the TPS for the training data, the difference ranging from about €20,000 for LoMST-CUSUM to more than €100,000 for WHC-LOF. This is because the faults previously classified as FPs are now classified as TPs. For CCA and EDP, there is no effect because no extra TPs are captured. For the test data, altering the TP period from 2–60 days to 2–90 days only has a small effect. This is due to the fact that no new TPs have been captured. However, the formula

for calculating the TP savings has changed slightly due to the change from 60 to 90 days, increasing the TP savings slightly.

Altering the lower bound of the TP period from two to one days only has an effect on TPS for the CCA model for the training data. This is because three faults were predicted within one day of the annotated failure with the CCA. These faults were classified as FPs for a range of 2–60 days but as TPs for a range of 1–90 days. There is no effect for the test data.

Halving the replacement costs for each component leads to a large reduction in TPS for each model for the training period (compared to the original case). This is because the replacement costs dominate for this period and therefore have a large effect on the savings. This is not the case for the test period because of the low number of FNs. For the NBM-LI, NBM and EDP models, the savings are increased on the order of €10,000, and for the LoMST-CUSUM, the CCA and WHC-LOF, the savings are decreased by about €10,000. These differences are due to the different numbers of FNs and TPs, which are both affected by the replacement costs.

Removing inspection costs for repeat alarms only affects the TPS of the model LoMST-CUSUM for the training data because this is the only model that contained repeat alarms (for the generator bearing). Even then, the effect is fairly small (on the order of €10,000. For the test data, it only affects the TPS for the CCA model. In this case, the TPS is increased quite significantly (on the order of €50,000) because five repeat alarm predictions occurred.

In conclusion, the evaluation method assumptions can have a large effect on the results. The assumptions mainly affected all the models in the same way, meaning that the final choice of model remained the same, regardless of the assumptions. Further analysis with a longer test data period would be useful for understanding these effects in more detail.

### 4.2. Qualitative Evaluation of Each Method
#### 4.2.1. NBM

The NBM model generated alarms for most of the analysed failures. Nevertheless, since they were raised with higher anticipation, the evaluation criteria classified most of them as FPs. This highlights the difficulty in comparing prediction models that have different characteristics. One of the drawbacks of the NBM model is that it is unable to detect failures that happen suddenly (such as a sensor malfunction), being more targeted at detecting components' degradation over time. It did not identify the problem at the hydraulic group because it probably had poor correlation to the other available turbine sensors, leading to a prediction with increased uncertainty.

#### 4.2.2. LoMST-CUSUM

The LoMST-CUSUM has a high hit rate in most of the components. Because the method has an advantage in accumulating effectively small-magnitude early symptoms over time, it performs well at detecting wear-out component failures (i.e., due to a longtime running in poor working conditions) rather than the temporary and random type of failures. In some of the cases, i.e., using different data (turbines), the method produces too many FPs. This is the focal point that can be improved. Finding the right subset of signals is the key to detection.

#### 4.2.3. WHC-LOF

The WHC-LOF method was able to detect most of the failure events in this challenge. It has the advantage to predict unknown outliers and does not need a large dataset to train every specific failure because the algorithm is trained with only data considered normal. There is no time dependence because it is possible to find patterns by comparing the parameter from multiple turbines instead of using the time series of one or more parameters. This time independence was probably the success to detect the failure of the hydraulic group in the test period. This method has the disadvantage to be site-specific and the pre-processing analysis to identify the anomaly clusters is necessary for each group of turbines and failure type.

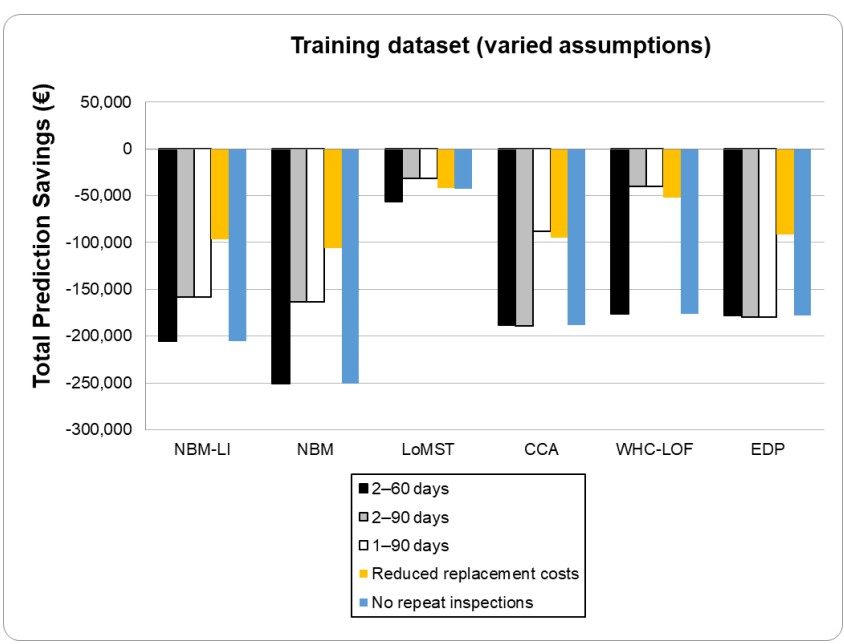

(**a**) Training period

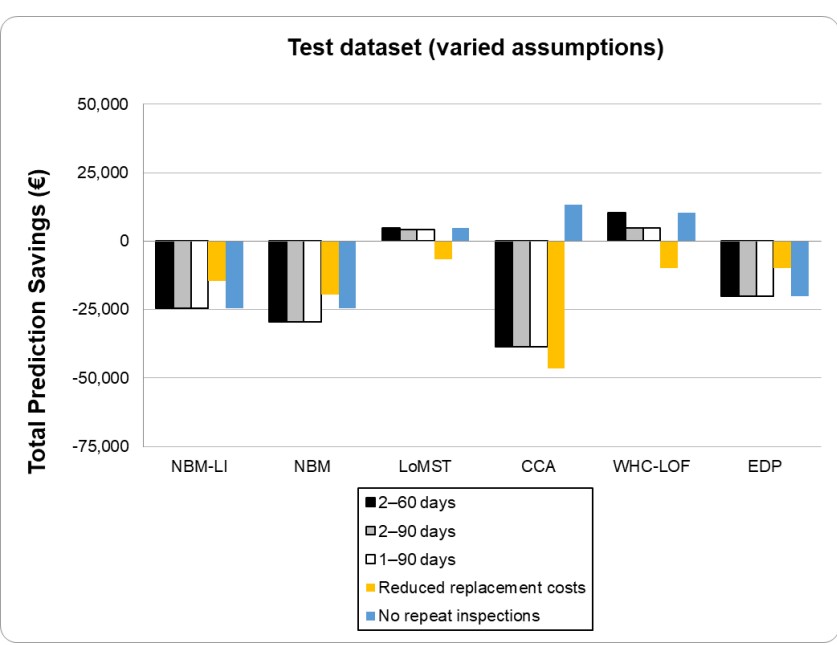

(**b**) Test period

**Figure 14.** Effect of variations on Total Prediction Savings (TPS) for each model.

### 4.2.4. NBM-LI

Introducing lagged variables increased the predictive capability of the model. This information from previous times can be informative by providing context to the current state of the turbine. For example, if there is a trend of decreasing rotor speed, this would mean that the rotor was previously running quickly, so we might expect hotter main bearing temperatures than in the case of an increasing rotor speed.

This method requires choosing which SCADA signals to predict the normal behaviour of, associating abnormalities with potential failures in an associated component. During analysis of the data set, errors in the predicted generator slip ring bearing temperature were found to be indicative of failures in the generator bearing, so the normal behaviour of this signal was associated with the health of this component.

### 4.2.5. CCA

The CCA method can detect faults of most components by training the model with normal data and describing the trend of the wind turbine operational state with statistics. When a fault occurs during the test, the statistics will exceed the threshold, so the model can identify the known and unknown wind turbine abnormal states. The selection of measurement variables and training samples has a great influence on the performance of the model, so it is necessary to select different variables and training samples for a large number of experiments. In addition, CCA is an anomaly detection method. In this challenge, the model is constructed for the whole wind turbine, so it is impossible to obtain specific fault types. It could be used to monitor specific component conditions in the future by modelling specific components separately.

### 4.2.6. KCPD

KCPD was able to demonstrate its abilities as a data pre-processing method by identifying change-points in SCADA signals that are caused by changes in the underlying data-generating process. The resulting benefits are threefold. Firstly, it enables clean training data for NBMs, a necessary precondition for the approach to work, through the exclusion of training periods containing change-points. Secondly, it adds information to the malfunctions annotated by operators, e.g., which signal and therefore sub-component was affected specifically by a certain maintenance action. In addition, lastly, it enables the data scientist to pose further specific inquiries (signal, component and time) regarding potential maintenance actions not reported. All in all, the method is a valuable addition to reduce ambiguities in real-world SCADA data processing.

### 4.3. Challenges of the Evaluation Method

One of the main challenges encountered in the evaluation process was quantifying the financial gain. When an algorithm raises an alarm and a team is sent to perform maintenance, it is impossible to know for sure how this failure mode would evolve if it had remained undetected, and therefore it is difficult to estimate the theoretical future maintenance/replacement cost and the achieved gain in detecting the failure early. In addition to this, it is difficult to estimate the lead-time when the algorithm raises an alarm, and therefore the success of a model depends highly upon the definition of a true positive.

Another source for evaluation-related challenges is the heterogeneity of the provided solutions. This becomes clear when comparing the rules the different algorithms apply to generate alarms (see Table 8). Formally, it can be difficult to directly compare models that generate different alarm KPIs (Key Performance Indicator), different outputs and formats. A more profound difficulty, however, arises from the potential multitude of hyperparameters involved in alarm generation for each algorithm. Every anomaly detection algorithm requires some threshold to distinguish between normal and abnormal conditions, and its choice usually depends on the the domain specific risks associated with false classifications [44]. In SCADA-based wind turbine monitoring, this manifests itself for example in averaging of anomaly metrics over time or specific rules, such as 'alarms on x consecutive days', to increase algorithm robustness against false alarms. The results, however, can be highly sensitive to the choice of such alarm generation thresholds and rules which complicates an effective evaluation and comparison between methods. As a starting point, we encourage reporting of performance metric sensitivity to hyperparameter choices rather than results for one specific setting only (threshold, averaging, etc...). Moreover, ML research has suggested evaluation metrics independent from specific threshold choices, such as the area under the receiver operating curve (AUROC or simply ROC, see e.g., [44,62,63]) and their adoption is therefore suggested in the future.

**Table 8.** Overview of data post-processing approaches by solution.

| Solution | Alarm KPI | Temporal Resolution | Threshold | Remark |
|---|---|---|---|---|
| NBM | Model error | 24 h | $+/- 3$std (training) | Alarm if on >3 out of last 7 days |
| LoMST-CUSUM | Cost function | 1 h | Different by component | Empirical from training data |
| WHC-LOF | Cumulative | 1 week | >100 | total anomaly per week |
| NBM-LI | Model error | 10 min | $+/- 15$std (training) | All anomalies raised alarms |
| CCA | SPE | 10 min | 13.42 | – |
| KCPD | Cost function | 24 h | 80 | Empirical from different external SCADA data sets |

*4.4. Evaluation of the Collaboration Method*

The collaboration method applied in this work resulted in the successful creation of six new fault detection models for the challenge providers to use. A total of 80 people from 26 different countries signed up to participate, with a diverse range of backgrounds and experiences. We carried out a total eight different workshops attended by different participants. There was an active exchange of ideas on the digital platform. The people who did not contribute a solution had access to all the discussions and results, and ultimately benefited from the process as well.

The benefits of the method can be summarised as follows:

- EDP received six new solutions to their challenge, two of which performed significantly better than their own method for the provided datasets. The average performance of all solutions was slightly better than the EDP method;
- EDP obtained access to the knowledge and code exchanged during the workshops and on the digital platform, as well as to the people participating. They were able to further their understanding on the topics of fault detection, data pre-processing and model evaluation;
- The monthly meetings combined with the digital platform provided an excellent opportunity for participants to exchange ideas and knowledge, as well as to ensure continued motivation and guidance;
- A range of people with different backgrounds got access to the challenge, leading to a large diversity of solutions and to some interesting exchanges, which would not have otherwise happened;
- The participants got to apply their methods to measurement data from a real wind farm under real conditions in collaboration with a real customer;
- The participants learned the difference between theoretical studies and real studies together with customers, when the required data are not always available in exactly the required format or volume.
- All the participants received access to the documentation of the workshops and the all of the knowledge related to the topic shared within the project;
- All the participants made new contacts and connections;
- Both EDP and the participants had the opportunity to discuss and test various evaluation methods.

The challenges related to the method include:

- The digital platform requires further functionalities, such as automatic notifications and regular summaries, in order to improve activity;
- It is important for the ecosystem operators to ensure that the challenge provider remains fully engaged throughout the project;
- Further datasets over longer time periods and including more faults would improve the evaluation process;

- More information about the actual maintenance activities that took place in the turbine, with information such as what was done (component fixed or replaced?) and the associated cost would be useful in the future;
- A pre-defined evaluation method would help direct the efforts more clearly from the start;
- A co-innovation process allowing different solutions to be combined may improve the results even more;
- A more formally-defined set of workshops including pre-defined goals and steps for each workshop would help the co-innovation effort;
- Definition of standard data formats or even the provision of a standardised docker for uploading code would reduce the evaluation effort and make the results more accessible to the challenge providers;
- Some broader challenges related to data sharing and co-innovation that have been highlighted during this work need to be solved (see Section ).

The combination of these benefits and challenges can be used for developing future co-innovation processes.

## 5. Impact of the Results on the Wider Community

In this section, the impact of the results of this work on the wider community is discussed, firstly in terms of applying the developed algorithms in practice and then in terms of the general applicability of the collaboration method.

### 5.1. Application of the Algorithms

Data driven O&M applications in the wind energy domain have followed the developments in the field of ML. In terms of model classes and architectures, we have seen a move from classical, shallow algorithms to deeper and more complex architectures. Conceptually, there has been an extension of classical NBM approaches towards combining methods beyond supervised learning and incorporating the latest results from anomaly detection research. This is also reflected by the variety of contributions within this paper.

However, in practice, we often see that the adoption of new approaches severely lags the development in academia. Many wind farm owner/operators currently limit their analysis to rather simple approaches, with real-time decision support only implemented at a very basic level.

We think that there are several reasons for this—and many of them can be addressed with the help of collaborative, digital ecosystems. Firstly, many studies focus on a relatively small, homogeneous database while in practice operators have large, diverse portfolios to manage. The digital ecosystem can help mitigate this effect by accumulating data sets from multiple data providers and making them available in a unified format, providing a more realistic environment for developing and benchmarking new methods. Additionally, this enables testing on how well the developed methods scale to a large database, an integral requirement in practice. Other major challenges are related to data quality and model transparency. While the former is widely known [61] and the wind energy community has started to address them (e.g., [59]), the latter has rarely been addressed so far in wind specific applications, despite the fact that eXplainable AI (XAI) has developed into a major subfield in ML research [64]. Digital ecosystems can help to overcome these hurdles, for example by setting challenges specifically targeted to these issues or enforcing transparency and robustness requirements for challenge solutions. We think that it is important to address these issues as a research community. Otherwise, there is a risk that further developments, e.g., towards the latest, even more complex ML architectures will (again) show impressive results but struggle with adoption in practice. Moreover, they can bring together researchers and operators directly to discuss the practical issues of specific solutions as well as develop the most relevant research questions in collaboration.

Finally, an open data mindset is required to effectively bring the latest generation of ML methods, transformer architectures, to the wind energy domain. These models

are not only characterised by their outstanding performance on many computer vision and Natural Language Processing (NLP) tasks; they also require massive amounts of data to be trained. While transfer learning approaches allow the use of pre-trained models, such as BERT, GPT or T5 [65], significant amounts of domain specific, labelled data are still required to fine-tune them. Moreover, this data might be of different nature than the standard wind turbine SCADA data sets often shared today. Transformers originated in NLP, and therefore would be the architecture of choice for any NLP-related tasks in the wind energy domain. With respect to condition monitoring, these could include knowledge extraction from maintenance reports or the generation of descriptive text based on model results. Initial steps in this direction have been taken by [66], where transformer models are used to first generate event descriptions based on SCADA sensor patterns and then to select matching maintenance actions from a lookup-table. Further developments in this area, however, depend heavily on the availability of domain specific, labelled text documents. To date, such data can hardly be found open source due to the prevailing confidentiality concerns of data owners and the cost of domain expert annotations. Collaborative digital ecosystems could act as trusted mediators between data owners and users, potentially applying some data anonymisation techniques [67] and thereby facilitating the transfer of latest ML methods to the wind energy domain. Furthermore, they could be used to develop common solutions against adversarial attacks [68] as well as for ensuring data security and privacy when used for real-time decision support.

### 5.2. Application of the Collaboration Method

The challenge-based digital ecosystem introduced in this paper has a high potential to be used by the wider community (within and beyond wind energy), As well as providing realistic environments for developing and benchmarking new ML methods and testing how well the developed ML methods scale to a large database, this could include providing "safe spaces" to share data and knowledge on particular topics or surrounding particular open data sets, making key challenges from different stakeholders easily accessible to a diverse range of people with varying experience and perspectives, providing a central location for documenting information and sharing knowledge, bringing together students and companies from all over the world and developing community-based data standards and data privacy solutions.

In order to do this, several broader challenges need solving. These include:

- A general fear of losing competitive advantage or communicating a negative message related to data sharing and open innovation that still exists in some organisations or entire industries [69];
- A lack of organisational structures that accommodate co-innovation or data sharing in some industries (e.g., [70]);
- The traditional approach of "silos" learned by the current workforce both through their education [71] and through their organisation [72];
- The issue of data privacy and security related to sharing commercially sensitive data;
- A general "black and white" thinking of many people and organisations when it comes to data and knowledge sharing, where "black" refers to "sharing everything with the public" and "white" refers to "sharing nothing". There is a large grey area that can be exploited to the benefit of everyone.

In general, this paper aims to present a single example of a bottom-up approach that will gain momentum and grow as more and more people and organisations get brave enough to test and try out new co-innovation approaches. The more the benefits can be highlighted and the challenges can be worked on together, the more success the approach will have and the more open people and organisations will become.

## 6. Conclusions

A literature review on the challenges related to implementation of digitalisation in the wind energy industry showed that there is a strong need for new solutions that enable co-innovation within and between organisations.

Therefore, a new collaboration method based on a digital ecosystem was developed and demonstrated. The method incentivises data sharing and allows a fair evaluation of solutions, makes wind energy data FAIR, provides a central location for data and knowledge related to a certain topic within the sector, includes solutions and code for data filtering and standard analysis tasks, and allows data standards and data structure translation solutions to be published and shared.

The ecosystem is centred around specific industry-relevant "challenges", which are defined by "challenge providers" within a topical "space" and made available to participants of the ecosystem via the digital platform. The data required in order to solve a particular "challenge" is provided by the "challenge providers" under the confidentiality conditions they specify. This can include only allowing specific people to access their space, requiring them to sign agreements or preparing the data so that it is anonymous or normalised. A "challenge" is defined as a fixed problem with a motivation, goal, expected outcome and deadline.

The method was demonstrated via a case study, the EDP Wind Turbine Fault Detection Challenge. The aim of this challenge was to identify failures in five of the major Wind Turbine components and advise an intervention to the wind farm operators in order to reduce corrective maintenance costs. The collaboration method was applied via a dedicated space created on the digital platform. The ecosystem owners supported the challenge providers by coordinating the entire challenge process, including the acquisition of participants, moderating and documenting workshops, offering support using the digital platform, sending regular email updates and providing a downloadable docker for beginners.

Six solutions using Normal Behaviour Models, Combined Local Minimum Spanning Tree and Cumulative Sum of Multivariate Time Series Data, Combined Ward Hierarchical Clustering and Novelty Detection with Local Outlier Factor, Normal Behaviour Model with Lagged Inputs, Canonical Correlation Analysis and Kernel Change-Point Detection were submitted to this challenge. Evaluation of the results showed several advantages and disadvantages of the different methods. Two of the methods performed significantly better than EDP's existing method in terms of Total Prediction Costs (order of €120,000), and the average of all the solutions was slightly better (order of €2000). During the evaluation process, several challenges were experienced relating to the heterogeneity of the provided solutions, such as different alarm KPIs, different outputs and different formats.

The case study demonstrated that the digital ecosystem is a promising solution for enabling co-creation in wind energy. It provided a number of benefits for both challenge and solution providers, including access to data, code, knowledge and people skills. Future improvements being developed include more formal evaluation methods, digital platform notifications as well as standardised data and data structures for improved evaluation and access to the results.

Finally, the paper allowed the potential and the challenges related to co-innovation and data sharing in general to be identified and discussed. The work provides a basis and an inspiration for future studies and initiatives attempting to improve collaboration within and across industries in general.

**Author Contributions:** Conceptualization, S.B.; methodology, S.B.; software, L.A.M.L., Y.S., J.Q., E.L., Y.L., S.L., X.Z. and F.H.; validation, L.A.M.L., Y.S., J.Q., E.L., Y.L., S.L., X.Z. and F.H.; formal analysis, L.A.M.L., Y.S., J.Q., E.L., Y.L., R.F., S.L., X.Z. and F.H.; investigation, S.B., L.A.M.L., Y.S., J.Q., E.L., Y.L., S.L., X.Z. and F.H.; resources, L.A.M.L., Y.S., J.Q., E.L., Y.L., R.F., S.L., X.Z. and F.H.; data curation, L.A.M.L., Y.S., J.Q., E.L., Y.L., S.L., X.Z. and F.H.; writing—original draft preparation, S.B., L.A.M.L., Y.S., J.Q., E.L., Y.L., S.L., X.Z. and F.H.; writing—review and editing, S.B.; visualization, S.B., L.A.M.L., Y.S., J.Q., E.L., Y.L., S.L., X.Z. and F.H.; supervision, S.B.; project administration, S.B.; funding acquisition, S.B. All authors have read and agreed to the published version of the manuscript.

**Funding:** E.L.'s research is supported by a Fulbright Scholarship in collaboration with the Indonesian Government (DIKTI-Funded Fulbright). The APC was funded by OST. R.M.G.F.'s research was partially supported by the European Union Horizon 2020 fund through the WATEREYE project (Grant No. 851207) and by the Research Council of Norway through the AIMWIND project (Grant No. 312486).

**Institutional Review Board Statement:** Not applicable.

**Informed Consent Statement:** Not applicable.

**Data Availability Statement:** The data used for this challenge were provided by EDP on their Open Data platform: https://opendata.edp.com/pages/homepage/; accessed on 25 July 2022.

**Acknowledgments:** Many thanks to EDP, and in particular Mafalda Magro, Hugo Virgos and Sofia Ganihla, for working together with us on the challenge and for providing the data sets. We are also grateful to Charlie Henderson and Stacker Group for their IT support and development work on the underlying digital platform, Relight.

**Conflicts of Interest:** The authors declare no conflict of interest.

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
