# Peer review of "Enabling Co-Innovation for a Successful Digital Transformation in Wind Energy Using a New Digital Ecosystem and a Fault Detection Case Study"

_energies, doi:10.3390/en15155638_

Round 1
Reviewer 1 Report
Thank you for addressing the comments of this reviewer. The paper is now much more comprehensive - no further changes are needed in the opinion of this reviewer.
Author Response
Thanks for the review!
Reviewer 2 Report
The authors present a very good and interested paper.
No comments.
Author Response
Thanks for the review!
Reviewer 3 Report
In my opinion, the article is interesting and very well written, I only have a few minor comments: 1. Please consider separating from the "introduction" section a short "literature review" section. As it stands, the introduction section is very long. 2. Figure 1 is somewhat unreadable. Moreover, the information contained in "green circles" may be worth presenting in a different form? Show their mutual relations? 3. Use the latest literature on the subject. It is worth referring to the following articles: Liang, J., Irfan, M., Ikram, M., & Zimon, D. (2022). Evaluating natural resources volatility in an emerging economy: The influence of solar energy development barriers. Resources Policy, 78, 102858. Nasr, A. K., Kashan, M. K., Maleki, A., Jafari, N., & Hashemi, H. (2020). Assessment of barriers to renewable energy development using stakeholders approach. Entrepreneurship and Sustainability Issues, 7 (3), 2526. etc. 4. The article is quite extensive. Perhaps some parts of it could be included as "a.ttachments"? Good luck!
Author Response
Thank you for carrying out this review. Our response to your comments are below and the changes we made to the paper are marked in red in the marked-up version. We hope our answers are satisfactory.
In my opinion, the article is interesting and very well written, I only have a few minor comments:
- Please consider separating from the "introduction" section a short "literature review" section. As it stands, the introduction section is very long.
Thank you for this comment. We have implemented this change as suggested.
- Figure 1 is somewhat unreadable. Moreover, the information contained in "green circles" may be worth presenting in a different form? Show their mutual relations?
We have reviewed this point and think this is due to the quality and size of the figure. It seems to be more readable now. Also, we will ask the editors if it's possible to use the entire page width.
- Use the latest literature on the subject. It is worth referring to the following articles: Liang, J., Irfan, M., Ikram, M., & Zimon, D. (2022). Evaluating natural resources volatility in an emerging economy: The influence of solar energy development barriers. Resources Policy, 78, 102858. Nasr, A. K., Kashan, M. K., Maleki, A., Jafari, N., & Hashemi, H. (2020). Assessment of barriers to renewable energy development using stakeholders approach. Entrepreneurship and Sustainability Issues, 7 (3), 2526. etc.
These papers are certainly interesting. We have included them both in Section 2.2.
- The article is quite extensive. Perhaps some parts of it could be included as "a.ttachments"? Good luck!
We have considered this suggestion but cannot find a good solution that keeps the story of the paper complete, so we have not done this.
This manuscript is a resubmission of an earlier submission. The following is a list of the peer review reports and author responses from that submission.
Round 1
Reviewer 1 Report
There are some minor faults:
-line 314.....Figure ??--complete the number of the figure, please
-line 317....the same error for the Figure ??
-pg. 11--Table 2....the indexes: rpl, rpr, insp...means ???.I suggest put them as references, like for Table 3
Reviewer 2 Report
In the paper the authors describe the results of research conducted under two grants (EU - Horizon 2020 and the Norwegian Research Council). Contrary to what has been announced, we are not dealing here with a thoroughly prepared literature review, and the conclusions regarding the implementation of digitization in wind energy are obvious. In fact, the authors want to justify the creation of the WeDoWind Ecosystem. The presented analysis of case study the "EDP Wind Turbine Fault Detection Challenge" based on simple profit estimation methods justifies the correctness of the WeDoWind Ecosystem operation. However, the obtained results are obvious, more useful for lower-level personnel. Taking into account the integration of complex diagnostic systems, no synergy effects have been demonstrated that would contribute to further research on the problem of efficient exploitation of wind turbines. It is worth mentioning that the article does not discuss the WeDoWind Ecosystem in a structural way, but only tries to show the positive aspects of its application through the analysis of this case study. The promotion of the WEDoWind Ecosystem software presented in this way is a premise for a negative assessment of the paper and its non-approval for publication.
Reviewer 3 Report
This paper introduced an interesting platform, WeDoWind, for connecting challenge providers and solution providers for wind energy systems. However, it looks more like an initiative proposal rather than a research paper. The contribution (e.g., scientific and technologic innovations, new insights and knowledge) of the paper is not clear. The case study is very detailed but its dependency on the WeDoWind platform seems weak for proving the effectiveness and innovation of the platform. Hence, the paper may not be suitable for publication in the journal in the current form. The authors may consider more relevant journals/conferences.
Reviewer 4 Report
An interesting paper that focuses on a very emerging area (digitalisation in the wind industry). The paper focuses on providing a holistic review of the challenges in data-driven decision making in the wind industry and also explores few notable solutions (from the EDP wind turbine fault prediction challenge). The paper is nicely written, but there are a few areas which need some additional clarification and explanation to make it clearer and comprehensive for the average reader of this journal:-
- This reviewer seems to find that there is somewhat of a disconnect between the FAIR framework and the idea of the WeDoWind Ecosystem discussed in this paper. It is not clear what the paper exactly aims to focus on from the introduction - is it mainly the EDP challenge algorithms, or more about the FAIR data policies and practices, or rather about the complete ecosystem for O&M in the wind industry? It would be useful to make this clearer in the abstract and the introduction with a sentence or two.
- The paper discusses the FAIR framework for overcoming challenges in data sharing and adoption in the wind industry. However, the discussion surrounding data quality is very limited in this paper. As FAIR cannot alone accomplish digitalisation without without an equal focus on both data quantity + quality - it would be useful it the paper can provide some more context on the challenges in data quality in the wind industry. There is lack of unified standards and inconsistency in datasets, besides several other issues which past publications (e.g. "Long-term research challenges in wind energy – a research agenda by the European Academy of Wind Energy") have highlighted in this domain. The paper does not cover such challenges pertaining to data quality at a sufficient level of detail. Additionally, what are the challenges in using e.g. the 6 models discussed in real-time decision support? Are there issues in data security and privacy for usage of technologies like Internet of Things, communication challenges, cloud storage and computational constraints etc. before the discussed AI algorithms can deliver value in real-world and real-time decision making?
- While the paper reviews the EDP challenge solutions, there is little to no mention of the limitations in the algorithms used in the challenge (and in general in the wind industry). While an off-the-shelve AI model can be leverage with wind turbine data either for a classification, regression or optimisation task in the wind industry - how can transparency and explainability in the algorithms play a major role to instil trust and confidence for real-world utilisation of the algorithms? Past publications e.g. "A Combined Algorithm for Data Cleaning of Wind Power Scatter Diagram Considering Actual Engineering Characteristics" highlight specialised algorithms that can be used to make data-driven predictive models more robust, and e.g. "Scientometric review of artificial intelligence for operations & maintenance of wind turbines: The past, present and future" have also highlighted the pressing needs to transition to Explainable AI methods beyond high predictive accuracy. It would be useful if this paper can enunciate briefly on some of these challenges (that have already been discussed in the past but not outlined in the paper) towards achieving a "FAIR" ecosystem that encourages data usage in more robust applications of modern AI/ML algorithms.
- It would be very useful if the paper can briefly comment on the risks of adversarial attacks in the 6 algorithms reviewed and beyond that, in general - the risks of adversarial attacks when aspiring for a FAIR framework.
- Based on the literature review conducted across the 6 algorithms and in general based on the current use-cases of the FAIR framework - it would be useful if the paper can comment on whether the wind industry should envision for a future ecosystem for digitalisation in O&M based on more traditional AI algorithms (e.g. Normal Behaviour Models), or more sophisticated (and currently hyped) deep learners (e.g. GPT, BERT, Reformers, Transformers etc.). Which of these directions look more promising based on the EDP challenge outcomes in the opinion of the authors?
- A Discussion section could be incorporated before the conclusions to enunciate on the strengths and limitations of the FAIR framework (and in general) the practice of carrying out competitive challenges in the wind industry such as hackathons. Besides the advantages, are there any risks in sharing commercially sensitive data with the public? The paper mentions brief about anonymisation in few sections, but it would be useful to bring together the key-takeaways from the paper (rather than just from the bullet points summarising the take-aways from the EDP challenge). What can the average reader in this journal learn from the FAIR framework - challenges in digitalisation and transition to AI/ML - real-time decision support, explainability and transparency, focus beyond cost savings, potential for a comprehensive O&M solution that combines e.g. computer vision techniques (drones for inspection) with more traditional SCADA data streams to provide an all round FAIR solution etc. How can the wind industry potentially ensure that data across different aspects e.g. drone images, SCADA, alarm records, O&M manuals etc. be better integrated to facilitate a full-fledged robust ecosystem closer to real-world utilisation by wind farm operators?